# Response of Export Production and Dissolved Oxygen Concentrations in Oxygen Minimum Zones to pCO$_2$ and Temperature Stabilization Scenarios in the Biogeochemical Model HAMOCC 2.0

**T. Beaty[1,2], C. Heinze[3,4], T. Hughlett[5], and A. M. E. Winguth[5]**

[1]{Northern New Mexico College, Española, New Mexico}

[2]{New Mexico Consortium-Biological Laboratory, Los Alamos, New Mexico}

[3]{Geophysical Institute, University of Bergen, and Bjerknes Centre for Climate Research, Bergen, Norway}

[4]{Uni Research Climate, Bergen, Norway}

[5]{Department of Earth and Environmental Sciences, University of Texas Arlington, Arlington, Texas}

Correspondence to: T. Beaty (tbeaty@newmexicoconsortium.org)

## Abstract

Dissolved oxygen (DO) concentration in the ocean is an important component of marine biogeochemical cycles and will be greatly altered as climate change persists. In this study a global oceanic carbon cycle model (HAMOCC 2.0) is used to address how mechanisms of oxygen minimum zones (OMZ) expansion respond to changes in CO$_2$ radiative forcing. Atmospheric pCO$_2$ is increased at a rate of 1% annually and the model is stabilized at 2 X, 4 X, 6 X, and 8 X preindustrial pCO$_2$ levels. With an increase in CO$_2$ radiative forcing, the OMZ in the Pacific Ocean is controlled largely by changes in particulate organic carbon (POC) export, resulting in increased remineralization and thus expanding the oxygen minimum zones within the tropical Pacific Ocean. A potential decline in primary producers in the future as a result of environmental stress due to ocean warming and acidification could lead to a substantial reduction of POC export production, vertical POC flux, and thus

increased DO concentration particularly in the Pacific Ocean at a depth of 600-800 m. In contrast, the vertical expansion of the OMZs within the Atlantic is linked to increases POC flux as well as changes in oxygen solubility with increasing seawater temperature. Changes in total organic carbon and increase SST also lead to the formation of a new OMZ in the western sub-tropical Pacific Ocean. The development of the new OMZ results in dissolved oxygen concentration of $\leq 50$ µmol $kg^{-1}$ throughout the equatorial Pacific Ocean at four times preindustrial $pCO_2$. Total ocean volume with dissolved oxygen concentrations of $\leq 50$ µmol $kg^{-1}$ increases by 2.4%, 5.0%, and 10.5% for the 2 X, 4X, and 8 X $CO_2$ simulations, respectively.

## 1   Introduction

Rapid increases in concentrations of greenhouse gases ($CO_2$, $CH_4$, and $N_2O$) in the atmosphere since the 18[th] century have led to greenhouse gas radiative forcing and temperature change of 0.068 °C $dec^{-1}$ (Karl et al. 2015). Atmospheric $CO_2$ concentrations are predicted to continue to rise from the pre-industrial level of 280 ppmv up to ~800 ppmv by the year 2100 (IPCC 2013) or ~2000 ppmv by year 2400 under the assumption that all fossil fuel reservoirs are emitted into the atmosphere (Caldeira and Wickett 2003, Zachos et al. 2008). The anthropogenic $CO_2$ will be partially sequestered by the ocean and by the biosphere on time scales on the order of $10^4$ years. A rise in ocean temperature decreases the solubility of $CO_2$ in seawater and thus the $CO_2$ uptake into the ocean. In addition, the ocean buffer capacity decreases with rising $pCO_2$.

Changes in climate as a result of $CO_2$ emission will affect the oxygen distribution in the ocean. DO (dissolved oxygen) concentration in the ocean is affected not only by changes in ocean ventilation but also by solubility and the biological pump (Volk and Hoffert 1985). The biological pump is controlled by export production, vertical carbon flux and decay of particulate organic carbon, dissolved organic carbon and by the transport of biogeochemical tracers by the ocean circulation. Variations in seasonal and long-term DO concentration have been observed in sub-polar and subtropical regions (Whitney et al. 2007, Stramma et al. 2008). Climate models predict that DO concentrations in the ocean will continue to decline with the warming of the deep-sea due to the subsequent decline in solubility as well as variations in the biological pump due to changes in mixing and enhanced ocean stratification. The decrease of the DO concentration will likely result in the expansion of oxygen minimum

zones (Sarmiento and Orr 1991, Sarmiento et al. 1998, Schmittner et al. 2008, Shaffer et al. 2009) and a significant expansion of bottom water hypoxia ($<10$ μmol $O_2$ $kg^{-1}$).

There are five major non-seasonal OMZs discussed in the current literature, which are the eastern sub-tropical North Pacific OMZ (~15°-25°N), the eastern tropical Pacific OMZ (equatorial region), the eastern South Pacific OMZ (~15°-40°S), the Arabian Sea, the Bay of Bengal (Kamykowski and Zentara 1990, Karstensen et al. 2008, Paulmier et al. 2011), and one low oxygen zone (LOZ) or seasonal OMZ in the equatorial Atlantic. There is limited literature discussing the variability of the Atlantic and Indian Ocean OMZs; however, areas of the eastern North Atlantic OMZ are hypoxic with DO concentrations ranging from 40 to $<2$ μmol $kg^{-1}$ (Stramma et al. 2009, Karstensen et al. 2015). Pacific OMZs have been discussed extensively and there is strong evidence that expansion is already occurring (Oschlies et al. 2008, Stramma et al. 2008, Keeling et al. 2010, Stramma et al. 2012). An expansion of the OMZ, a shoaling of the depth of hypoxia (DOH; shallowest depth at which OMZ criteria is met), or a shoaling of the OMZ cores into the photic zone could have severe impacts most notably the decline in ecosystems in the ocean.

In this study, the core of the OMZ is defined as a dissolved oxygen concentration of $\leq 20$ μmol $kg^{-1}$ $O_2$ consistent with Helly and Levin 2004, Fuenzalida et al. 2009 and Paulmier et al. 2011. The OMZ boundaries are described to have a DO concentration of 50 μmol $kg^{-1}$. The maximum DO concentration of 50 μmol $kg^{-1}$ is more stringent than upper limits in other studies (Whitney et al. 2007, Karstensen et al. 2008); however, at these DO concentrations most microorganisms cannot survive (Kamykowski and Zentara 1990, Gray et al. 2002, Sarmiento and Gruber 2006, Paulmier et al. 2011) and therefore considered a reasonable criterion for non-seasonal OMZ. This study focuses on the extent of OMZ expansion and determining the relative strengths of two mechanisms of OMZ expansion, the export production and oxygen solubility.

## 2   Model Description

This study is conducted with the biogeochemical Hamburg Oceanic Carbon Cycle Model Version 2.0 (HAMOCC 2.0), which has been originally developed by Maier-Reimer and Hasselmann (1987) and Maier-Reimer (1993), and expanded to include an iron cycle, sedimentary phosphorus cycle, and improved atmospheric dust parameterization (Palastanga et al. 2011, Palastanga et al. 2013). The model utilizes an E-grid (Arakawa and Lamb 1977)

and has a horizontal resolution of ~3.5° x 3.5° with grid points 1.25° north and south of the
equator to resolve the equatorial upwelling belt. The model contains 11 layers (centered at
25,75,150, 250, 450, 700, 1000, 2000, 3000, 4000, and 5000 meters) with a total depth of
5000 meters (Heinze et al. 1999, Heinze et al. 2006, Heinze et al., 2009). HAMOCC 2.0
includes a sediment module with porewater and solid components that are coupled by a
reaction rate. The sediment module includes one 10 cm thick layer of bioturbated sediment,
which is further divided into 11 sub-layers. A more detailed description of the sediment
module can be found elsewhere (Heinze et al. 1991, Heinze et al. 1999, Heinze 2004).
The annually averaged version is computationally very economical and suitable for long-term
carbon cycle simulations of several 10,000 years. Long-term integrations are possible with
HAMOCC because of it coarse temporal and spatial resolution and because of the
computational efficient solution tracer equations by an upstream formulations (Maier-Reimer
and Hasselmann, 1987, Heinze and Maier-Reimer, 1999) that uses the prescribed annual
average circulation and hydrography of the Large Scale Geostrophic (LSG) ocean general
circulation model (Maier-Reimer et al., 1993; Winguth et al., 1999).
Atmospheric $CO_2$ and $O_2$ are exchanged between the ocean surface (top 50 m) and zonally
mixed atmospheric boxes. The air-sea gas exchange of $CO_2$ is determined by the difference in
the partial pressure of $CO_2$ in the sea surface and the atmospheric $pCO_2$, the gas transfer
velocity, and the requirement for a full equilibration of the surface layer inorganic carbon
system. The gas exchange of oxygen is an order of magnitude faster than that of $CO_2$. Oxygen
exchange is carried out according to a fixed transfer velocity and is assumed to be at
equilibrium between the atmospheric layer and the surface water at the temperature and
salinity-dependent saturation level. The solubility of dissolved oxygen depends on
temperature, salinity and pressure (Weiss 1970). The $O_2$ flux into the atmosphere is neglected
since the atmospheric concentration of $O_2$ is by far larger than the DO concentration at the
ocean surface.
The temperature-dependent annual export production of particulate organic carbon (POC) and
opal from the euphotic zone is calculated via Michaelis-Menten kinetics (Parsons and
Takahashi 1973) and $CaCO_3$ production is dependent on the particulate organic and opal
production. This relationship is based on the assumption that in the present day ocean there is
a dominance of the silicate producers (e.g. diatoms) over the calcareous plankton (e.g.
coccolithophores) (Falkowski et al. 2007). The POC export from the surface into the deep sea
is determined from organic carbon production in the uppermost layer and then transported to
the deep with a uniform sinking rate of 120 m day$^{-1}$. Remineralization of organic matter
depends on the availability of oxygen for consumption in the water column. Remineralization
of POC occurs as long as dissolved $O_2$ is larger than the minimum $O_2$ concentration $[O_{2min}]$
$= 10^{-5}$ mol L$^{-1}$ for bacterial decomposition of POC. A more detailed description of the model
can be found elsewhere (Maier-Reimer and Hasselmann 1987, Heinze et al. 1991, Maier-
Reimer and Heinze 1999, Heinze et al. 1999, Palastanga et al. 2011, Palastanga et al. 2013,
Beaty-Sykes 2014).

## 3 Experimental Design

The annually averaged version of the model was integrated to quasi-equilibrium state (200
kyr) with a stable atmospheric $CO_2$ concentration of 279.78 ppmv. The reference experiment
and all OMZ sensitivity experiments are started from the near-equilibrium state and integrated
for 30,000 yrs. For the reference experiment, the model is forced with flow fields from a LSG
simulation. The globally averaged potential temperature and salinity are 3.78°C and 34.8 psu
respectively (Winguth et al. 1999).
Carbon cycle sensitivity experiments are conducted in three sets of scenarios. The first set of
scenarios consists of a perturbation of the atmospheric $CO_2$ concentration relative to
preindustrial atmospheric levels (pCO$_{2ref}$; PAL) of 2 X $CO_2$, 4 X $CO_2$, 6 X $CO_2$, and 8 X $CO_2$
to explore the sensitivity of distribution of dissolved oxygen concentration to rising
atmospheric pCO$_2$ level. In these simulations, all other boundary conditions and model
parameters are kept at preindustrial levels (Table 1). In a second set of experiments the pCO$_2$
levels are accompanied by the associated changes of temperature at the sea surface as well as
in the deep sea to investigate the response of the dissolved oxygen distribution to increases in
$CO_2$ radiative forcing. In a third set of experiments; POC is kept at preindustrial level to
explore the relative strength of loss of $O_2$ solubility and oxygen consumption by
remineralization. The preindustrial POC experiments are simulated with at atmospheric $CO_2$
concentrations of 2 X, 4 X and 8 X $CO_2$. Stabilization scenarios and brief descriptions are
listed in Table 2.
In all $CO_2$ perturbation scenarios atmospheric pCO$_2$ is increased from preindustrial levels by
1% each year (t) until the perturbed atmospheric pCO$_2$ (pCO$_{2pert}$) is stabilized at its maximum
level (pCO$_{2max}$) by
$$for\ pCO_{2} < pCO_{2_{max}}: pCO_{2_{pert}} = pCO_{2_{ref}}(1 + 0.01)^{t}$$
$$and\ for\ pCO_{2} \geq pCO_{2_{max}}: pCO_{2_{pert}} = pCO_{2_{max}}. \tag{1}$$
The 1% increase of atmospheric $CO_2$ concentration follows the IPCC (2013) business as
usual scenario and is stabilized after 70 years for doubling of preindustrial $pCO_2$ (see also
Winguth et al. 2005). The second set of carbon perturbation scenarios includes the feedback
of increasing seawater temperature due to rising atmospheric $pCO_2$ (Fig. 1). Temperature
increases as a function of the 1% increase per time step of atmospheric $pCO_2$ and is
determined using Eq. 2 from Hansen et al., (1988) for the radiative forcing of $CO_2$ with the
addition of a climate model sensitivity of $A_t$=0.6870.
$$\Delta T = A_t\ 6.3 \ln\left(\frac{pCO_2}{pCO_{2_{ref}}}\right) \tag{2}$$
Therefore a doubling of $pCO_2$ results in a homogeneous increase in temperature of ~3°C,
which is consistent with the estimate of Archer (2005) and Hansen et al. (1988). Note that this
enhanced sensitivity includes climate feedbacks whereas the direct $CO_2$ warming for 2 X $CO_2$
is ~1.2°C (Ruddiman 2001, Houghton 2004). The resultant temperature change of the ocean
for the doubling of $pCO_2$ for 2 X $CO_2$, 4 X $CO_2$, 6 X $CO_2$, and 8 X $CO_2$ is 2.8°C, 5.9°C,
8.7°C, and 11.5°C respectively (Fig. 1). The temperature change is applied at all depths of the
ocean. Solubility and chemical kinetic equilibrium constants of the carbon cycle are adjusted
to the changes in $pCO_2$ and temperature at each time step in the temperature feedback
experiments.
In addition to experiments with increased $pCO_2$ with and without radiative forcing a reduced
biology scenario is added in which primary productivity and export (Si, $CaCO_3$, and organic
carbon) is set to zero following the approach of Maier-Reimer et al. (1996). The reduced
biology scenario is simulated with preindustrial $pCO_2$ (279 ppmv; Table 2).
Four additional simulations were conducted in order to explore how DO concentrations in the
model respond to changes in ocean ventilation. Velocity variables w, v and u are reduced
uniformly over the ocean globally by 25%, 50%, 75% and 100%. Diffusion is not changed in
these experiments and remains at preindustrial reference simulation values.

# 4 Results

## 4.1 Reference simulation

The relevant results of the reference experiment will be briefly discussed in this section. Prescribed temperature and salinity taken from Winguth et al. (1999) are comparable to the observed data from the World Ocean Atlas 2013 (referred hereafter as WOA2013; Locarnini et al. 2013, Zweng et al. 2013) and to the simulations of Maier-Reimer (1993). Simulated seawater temperature, dissolved oxygen and salinity are comparable to the World Ocean Atlas 2013 at 3000 m depth. Due to the slow ventilation of the ocean the WOA2013 data at 3000 m is more representative of preindustrial conditions. Compared to WOA2013, cooler simulated temperatures are projected for the Bering Sea by the LSG, leading to greater $O_2$ solubility at the surface and therefore higher DO concentration than the corresponding data from WOA2013 (Garcia et al. 2013, Locarnini et al. 2013). This bias may be partially linked to the long-term warming trend over the last decades (IPCC, 2013). Dissolved inorganic carbon (DIC) at the surface is similar to the simulations of Maier-Reimer (1993) and the observations from the WOA2013 (Locarnini et al. 2013) with the exception of the Arctic region in which the reference experiment simulated DIC concentrations at approximately 150 umol kg$^{-1}$ less compared to corresponding values simulated by Maier-Reimer (1993). The decreased simulated DIC in the Arctic region of this preindustrial simulation could be due to the addition of dust fields (Mahowald et al. 2006) and Fe and P cycles (Palastanga et al. 2011, Palastanga et al. 2013). Simulated ocean oxygen concentrations are comparable to Maier-Reimer (1993) and the WOA2013. POC, $CaCO_3$, and opal export and sediment composition are comparable to Maier-Reimer (1993). However, the model does trend toward a slightly higher POC in the tropical latitudes compared to Sarmiento and Gruber (2006) who used the chlorophyll concentration and sea surface temperature based empirical algorithm of Dunne et al. (2005). This bias may be linked to overestimation of export production in HAMOCC 2.0 linked to nutrient trapping (Najjar et al. 1992) at the equator region of the Pacific Ocean (Fig. 2). In addition, HAMOCC 2.0 simulates a slightly elevated export of $CaCO_3$ and opal export compared to corresponding observed values inferred from $CaCO_3$:POC and opal:POC export ratios (Sarmiento and Gruber 2006).

Simulated DO distribution in the reference simulation represents all five major non-seasonal oxygen minimum zones of the Pacific Ocean and Indian Ocean and the seasonal OMZ or low oxygen zone (LOZ; defined as dissolved $[O_2] < 90$ µmol kg$^{-1}$) of the eastern South Atlantic

Ocean (Fig. 3). However, due to the course model grid, the eastern subtropical and tropical North Pacific OMZ as well as the OMZs in the Indian Ocean (Arabian Sea and Bay of Bengal) are not resolved individually. The LOZ of the eastern South Atlantic Ocean is simulated in the reference experiment with a OMZ core of ~17-19 $\mu$mol kg$^{-1}$ O$_2$ and therefore, following the OMZ definition proposed here, the LOZ of the Atlantic Ocean is simulated as a non-seasonal OMZ.

The simulation is generally agreeable with the extent and depth of the OMZs, and DO core concentration values of the observations (Fig. 3). A model-data bias of the OMZ exist in the North Pacific Ocean resulting in the simulated OMZ reaching too far westward with the western boundary near ~180°W. The OMZ is also simulated too deep with a maximum depth of approximately 2300m. The difference in horizontal extent between the model simulation and observed in the eastern North Pacific OMZ may be attributed to the non-consideration of seasonally variability in the simulation. For the sub-tropical South Atlantic Ocean, the simulated OMZ core is located in a water depth ranging from 300 to 700 meters; which is slightly shallower than the OMZ core in the Indian Ocean. The total ocean volume with DO concentration of ≤20 $\mu$mol kg$^{-1}$ is approximately 1.4%.

## 4.2 Sensitivity of simulated dissolved oxygen to a reduced ventilation and biological pump

In order to explore the importance of biological pump (soft tissue pump) to the distribution and concentration of dissolved oxygen globally in the ocean we performed experiments in which P$_{POC}$ remains at preindustrial levels and atmospheric CO$_2$ is increased by 2 X, 4 X and 8 X CO$_2$ as well as an extreme scenario in which all productivity is reduced to zero. This extreme simulation, referred hereafter as the reduced biology scenario, is similar to the "Kill Biology" experiment by Maier-Reimer et al. (1996). In this simulation the atmospheric pCO$_2$ is set to preindustrial levels, which is in contrast to a simulated exponential increase in atmospheric pCO$_2$ in response to the diminished export production in the study of Maier-Reimer et al. (1996).

Due to the reduced export production, the DIC concentrations increase at the ocean surface by >400 $\mu$mol kg$^{-1}$ and by >200 $\mu$mol kg$^{-1}$ in the intermediate and deep-water masses at mid-latitudes. This leads to a significant rise in total alkalinity by an average of 550 $\mu$eq kg$^{-1}$. As a result, the pH increases by an average of 0.7 units despite the loss of calcification and CaCO$_3$

burial. Note that weathering rates are kept at preindustrial conditions in all simulations. Dissolved oxygen increases by ~150 $\mu$mol kg$^{-1}$ in the deep-sea and ~200 $\mu$mol kg$^{-1}$ in the intermediate water masses. The dissolved oxygen gradient in this reduced biology scenario is controlled by the air-sea gas exchange of $O_2$ at the surface and by the temperature-dependent solubility of oxygen: not by the vertical POC flux, which is set by definition to zero to the "killed" productivity. Thus consumption of oxygen by decay of POC is also diminished.

Experiments were preformed to evaluate OMZ response to weakened ventilation (eg. vertical, zonal and meridional velocities) in the model. Ventilation is decreased by 25%, 50%, 75% and 100% (Fig. 4). With a 25% decrease the OMZ core (< 25 $\mu$mol kg$^{-1}$) the OMZ deepen in each ocean basin and expand horizontal only slightly. The OMZs continue to expand in the experiment with 50% reduction in ventilation. Although $P_{POC}$ is decreasing as expected with the reduction in ventilation, DOC increase with a loss of 25% and 50% leading to the expansion of the OMZs in these two simulation. However, in simulations with 75% or greater loss in ventilation the DO concentration within the OMZ increases (Fig. 4). The increase in DO concentration coincides with a loss of $P_{POC}$ as well as DOC globally and in equatorial region. Simulated dissolved oxygen concentrations in the deep sea increase in the model in each reduced ventilation scenario due to the convection of oxygen at the poles.

## 4.3  Model sensitivity to changes in oxygen solubility

Solubility is another control of OMZ expansion; therefore, to determine how the model represents the expansion of OMZs due to solubility in response to radiative forcing, $P_{POC}$ is held at preindustrial levels and atmospheric $CO_2$ is increased to 2 X, 4 X and 8 X preindustrial concentrations.  Oxygen solubility is dependent on salinity, pressure and temperature and is calculated using the equation presented by Weiss (1970) yielding an average change of ~0.3 ml L$^{-1}$ per doubling of pCO$_2$ with the most significant changes in the deep sea. The relative strength of solubility and $P_{POC}$ on OMZ expansion will be examined in the discussion.

## 4.4  Sensitivity of the OMZs and global dissolved oxygen concentrations to increased pCO$_2$ without radiative forcing

The increased pCO$_2$ simulations that do not include radiative forcing (temperature increase; Eq. 2) result in small increases of dissolved oxygen in the model at the ocean surface due to

the enhancement of primary productivity. The small increase in productivity results in
increased DO globally. There are only slight changes in the distributions of DO concentration
for these simulations as compared to the simulation that include radiative forcing (Fig. 5).
Therefore, in order to discuss future changes in the OMZs the following sections address the
expansion of each OMZ and OMZ core as well as the global change at 2 X, 4 X, 6 X, and 8 X
$CO_2$ simulations that include the temperature feedback.

### 7 4.5 Sensitivity of the oxygen minimum zones to $CO_2$ radiative forcing

In each of the scenarios that include radiative forcing, the simulated OMZs expand (Fig 6).
The results show the formation of a new OMZ core in the tropical western South Pacific
Ocean. There are significant changes in the distributions of DO concentrations in all
simulations.

### 12 4.5.1 Simulated OMZ expansion in the eastern tropical Pacific Ocean in
### 13 response to $CO_2$ radiative forcing

For the 2 X $CO_2$ experiment, the OMZ cores (dissolved $O_2$ concentration $\leq 20$ μmol kg$^{-1}$) of
the OMZ in the eastern North Pacific Ocean expands to 65°N compared to the extent to 35°N
of corresponding OMZ in the 1 X $CO_2$ scenario. This OMZ merges with that of the eastern
South Pacific OMZ at the equator and therefore is considered as a single OMZ, hereafter
referred to as the eastern Pacific OMZ (Fig. 6). At a depth of 450 m it extends northward
around the northern boundary of the North Pacific gyre with dissolved oxygen concentrations
of $\leq 20$ μmol $O_2$ kg$^{-1}$ in the Gulf of Alaska. The southern boundary of the eastern Pacific OMZ
is located near the coast of Northern Chile at approximately 30°S at 450 meters depth.
Compared to the reference simulation, the OMZ in the 2 X $CO_2$ experiment expands 200 km
further to the south. The OMZ western boundary increases by approximately 550 km to
150°E. The depth of hypoxia (DOH) is between 150-250 meters. The OMZ has a max depth
of 1900 meters, 200 meters deeper than the reference simulation (Fig. 6). The OMZ core
shoals to 380 meters; however, the bottom boundary of the OMZ core does not deepen in the
2 X $CO_2$ simulation. The lowest oxygen concentration in the OMZ core is 17 μmol $O_2$ kg$^{-1}$ in
this simulation (Fig. 7).
The horizontal extent of the OMZ in the 4 X $CO_2$ scenario is similar to the 2 X $CO_2$
experiment with the addition of all of the North Pacific outside of the North Pacific Gyre

having a dissolved oxygen concentration of $\leq 50$ $\mu$mol $L^{-1}$ at a depth of 450 meters (Fig. 6). The depth of hypoxia shoals vertically to between 75-150 m from the surface in the North Pacific Ocean and remains in a depth range of 150-250 m in the South Pacific Ocean. The maximum depth of the Pacific OMZ increases to 2000 m. For the 4 X $CO_2$ experiment, the OMZ core extends ~100 km west and deepens by 200 m compared to the 2 X $CO_2$ simulations. The depth of the OMZ core does not change in the 4 X $CO_2$ simulations compared to the 2 X $CO_2$ simulations; however, the minimum dissolved oxygen concentration decreases to 14 $\mu$mol $kg^{-1}$ (Fig. 7).

There is further extension of the OMZ core south to approximately 50°S (central coast of Chile) at 450 m depth in the 8 X $CO_2$ scenario relative to the 4 X $CO_2$ experiment (Fig. 6). The OMZ core, at a depth of ~2000 meters, does not shoal or deepen in the 6 X and 8 $XCO_2$ compared to the 4 X $CO_2$ experiment. In the 8 X $CO_2$ simulation, the core becomes hypoxic with a minimum dissolved oxygen concentration of $\leq 8$ $\mu$mol $kg^{-1}$. The 6 X $CO_2$ experiment results in a minimum dissolved oxygen concentration of ~12 $\mu$mol $kg^{-1}$ (Fig. 7).

### 4.5.2 Simulated OMZ expansion in the eastern tropical South Atlantic Ocean in response to $CO_2$ radiative forcing

The horizontal expansion of the OMZ in the eastern South Atlantic in the 2 X $CO_2$ simulation remains similar to the reference scenario with a southern boundary at approximately 25°S and extends northward along the west coast of Africa to the southern tip of Morocco to approximately 15°N (Fig. 6). The depth of hypoxia shoals from between 250-450 m in the reference experiment to 150-250 m. The maximum depth of OMZ increases by 100 m to 1200 m. In the eastern South Atlantic, the OMZ core in the 2 X $CO_2$ experiment expands relative to the reference experiment southward by 580 km to approximately 19°S and northward by 110 km (~1° northward propagation). In the 2 X $CO_2$ experiment, the OMZ core expends vertically; it shoals to 450 m and deepens to 915 m, which is 65 m deeper than the reference simulation. The minimum dissolved $O_2$ concentration is reduced by 1 $\mu$mol $kg^{-1}$ relative to the reference experiment to 17 $\mu$mol $O_2$ $kg^{-1}$ (Fig. 7).

Relative to the reference simulation, the 4 X $CO_2$ simulation results in insignificant horizontal expansion of the OMZ in the latitudinal direction (Fig. 6). The most notable area of expansion of the OMZ is in the southwest direction in which the southwestern boundary of the eastern South Atlantic OMZ extends to ~30°S and ~20°W. The maximum depth increases by an

additional 100 m to a depth of 1300 m. The OMZ core expands symmetrically in east-west direction, by about 100 km, encompassing the Gulf of Guinea. The vertical expansion of the OMZ core is negligible between the 2 X and 4 X $CO_2$ simulations; however, the strength of the core increases significantly with a minimum dissolved $O_2$ concentration of 12 µmol kg$^{-1}$ (Fig. 7).

Horizontal expansion of the eastern South Atlantic OMZ does not occur between the 4 X $CO_2$ simulation and the 6 X or 8 X $CO_2$ scenarios (Fig. 6). In the 6 X $CO_2$ scenario the horizontal extent of the eastern South Atlantic Ocean at 450 m depth is reduced from the 4 X $CO_2$ simulation, where as in the 8 X $CO_2$ simulation the horizontal area expands back to the extent of the 4 X $CO_2$ simulation. The depth of hypoxia remains between 150-250 m depth for both 6 X and 8 X $CO_2$ experiments. The maximum depth of the OMZ increases to 1500 m in the 8 X $CO_2$ simulation. The OMZ core deepens to 1050 m and shoals from the 6 X and 8 X $CO_2$ scenarios to 375 m. The minimum dissolved $O_2$ concentration remains at 12 µmol L$^{-1}$ for both the 6 X and 8 X $CO_2$ simulations (Fig. 7).

### 4.5.3 Simulated expansion of the OMZ in the tropical Indian Ocean in response to $CO_2$ radiative forcing.

The expansion of the OMZ in the Indian Ocean is limited at the western boundary by the east coast of Africa and the eastern boundary is constrained by the Indonesian archipelago. The Indian Ocean OMZ includes the poorly resolved Arabian Sea and the Gulf of Bengal, which is limited by the Indian subcontinent. Compared to the reference simulation, the OMZ extends southward to 10°S in the 2 X $CO_2$ simulation and deepens by 100 m to 1100 m (Fig. 6). The OMZ core does not expand horizontally but deepens to 900 meters and shoals by 50m to 225 meters. The minimum dissolved oxygen concentration is 10 µmol kg$^{-1}$ and remains the lowest concentration for each of the emissions scenario (Fig. 7).

In the 4 X, 6 X, and 8 X pCO$_2$ simulations the horizontal expansion in the Indian Ocean OMZ is insignificant but it deepens to 1300 m, 1400 m, 1700 m, respectively. For the 4 X $CO_2$ experiment the OMZ core expands in the western direction to 45°E and deepens by 100 m to 1000 m; however, the upper boundary of the OMZ remains unchanged. In the 8 X $CO_2$ simulation the core expands southward by 650 km to approximately 16°S and shoals to 100 m for both the 6 X and 8 X $CO_2$ scenarios; however, the lower boundary remains unchanged compared to the 4 X $CO_2$ experiment. The depth of hypoxia is located between 25 m and 75

m in the reference experiment and in all $CO_2$ emission scenarios. It is important to note that
due to complex climate variability and nutrient trapping the annual tracer distribution in the
Indian Ocean consist of large uncertainties and thus the model-data bias is generally high in
the region.

### 4.5.4 Simulated OMZ formation in the western tropical Pacific Ocean in response to $CO_2$ radiative forcing

An OMZ core (<20 μmol $L^{-1}$ $O_2$) is simulated in the western tropical Pacific Ocean (143E,
2N) near the Bismarck Sea (Fig. 6 and 8). This region is modeled as a low oxygen zone
(LOZ) in the reference simulation. For the 4 X $CO_2$ experiment, the OMZ develops in <2000
yr integration with a minimum dissolved oxygen concentration of 17 μmol $L^{-1}$. The upper
boundary of the OMZ core remains unchanged for all perturbation simulations compared to
the reference. However, the OMZ core deepens from 725 m at 3 X $CO_2$ to 1000 m for the 8 X
$CO_2$ simulation.

### 4.6 Export of particulate organic carbon and changes in global dissolved $O_2$ concentration in response to simulations with $CO_2$ radiative forcing

Simulated total POC production and export production of POC ($P_{POC}$) from the euphotic zone
into the deep sea increases predominantly near the equatorial Pacific with a rise in seawater
temperature in response to $CO_2$ radiative forcing (Fig. 2). $P_{POC}$ in the northern Indian and
western tropical Pacific decreases in response to enhanced $CO_2$ radiative forcing most likely
due to nutrient trapping in the eastern Pacific Ocean. Changes $P_{POC}$ in the east Atlantic Ocean
are insignificant.
Global DO concentration decreases most rapidly during the first 2000 years of integration in
each carbon perturbation simulation. The reduction in global dissolved oxygen concentration
continues on average 1500 years beyond the year in which the peak $pCO_2$ emission value is
reached. The total ocean area with a dissolved oxygen concentration of <50 μmol $kg^{-1}$
expands at approximately 2% per ~3°C increase in seawater temperature which corresponds
to a doubling of $pCO_2$. The total ocean volume at which the dissolved $O_2$ concentration is
<50 μmol $kg^{-1}$ increases by 10.5% in the 8 X $CO_2$ simulations. The increase in the ocean
volume of hypoxic water in to the photic zone is insignificant (< 0.3%) due to the equilibrium
of oxygen between the atmosphere and the surface layer in the model. However, an area of
hypoxia forms in the photic zone of the sub-tropical North Pacific Ocean with a dissolved $O_2$
concentration of less than 12 $\mu$mol kg$^{-1}$.

## 5   Discussion

In this study we investigate the expansion of OMZ in a biogeochemical model as a result of
seawater temperature increase in response to $CO_2$ radiative forcing and changes in $P_{POC}$. It is
important to note that changes in ocean stratification due to ocean temperature and density
changes are not simulated and held constant at preindustrial conditions to allow for the long-
term carbon cycle feedback and an integration time of 30 kyrs. The focus of this study is to
examine changes in OMZs due to changes in solubility and remineralization. Furthermore,
this study determines the relative strengths of these mechanisms of OMZ expansion in the
biogeochemical model. The expansion of OMZs in this study is the result of changes in
temperature-dependent productivity and changes in $O_2$ solubility. Consequently, the OMZ
expansion simulated may be modest due to no consideration of a weakened connection
between the OMZs and the ocean surface in the future (Glessmer et al. 2011). It has been
suggested that the depth and strength of the thermocline may influence OMZ expansion and
contraction (Deutsch et al. 2007). An increase of the thermocline in a warmer climate may
result in a contraction of the OMZs due to reduced oxidative demand in hypoxic waters.
However, this study assumes a constant thermocline depth, as the temperature increase is
uniform at all depths. Other assumptions in this study are a constant nutrient inventory and
Redfield ratio. Changes in the elemental stoichiometry (carbon overconsumption) due to
rising $pCO_2$ has been suggested as a possible mechanism of enhanced volume of suboxic
water in the ocean due to the respiration of increased organic carbon. (Oschlies et al. 2008,
Riebesell et al., 2007). Measurements of dissolved oxygen concentration in the suboxic
regions of the oceans are limited (Levitus et al. 2013, Locarnini et al. 2006); however, paleo-
records and climate models support the assumption that ocean anoxic events occur during
periods of high $pCO_2$ (Knoll et al., 1996; Falkowski et al. 2011). Furthermore, OMZs have
expanded and contracted during the glacial interglacial cycles (Galbraith et al. 2004) as well
as on shorter time scales in response to Dansgaard-Oeschger (D-O) events (Cannariato and
Kennett 1999).
In the ventilation scenarios the model responds as expected for the reductions in ocean
ventilation of 75% or greater. Figure 4 illustrates the dissolved oxygen response to a near

complete shutdown of ocean ventilation resulting in an increase in the vertical oxygen gradient relative to the reference scenario.  The increase in DO in the reduced ventilation simulations of greater than 75% are due to the reduced $P_{POC}$ and thus reduced remineralization and to the convection of DO to the deep sea at the poles. The expansion of OMZ cores at lower changes in ventilation (eg. 25% and 50%; Fig 4) may be due the increase of DOC both global and regionally. An increase in dissolved organic carbon results in more available DO for remineralization in the model; therefore, even with reduced $P_{POC}$ the model response to atmospheric perturbations with an expansion of OMZs with a 25% and 50% reduction in ventilation.

The comparison between the 25% and 50% reduced ventilation experiment and the 4 X $CO_2$ with radiative forcing simulation, specifically in the Pacific Ocean basin, indicates that the decrease of dissolved oxygen concentration in the model is strongly controlled by $P_{POC}$ and solubility as the expansion is greater due to the influence of these mechanisms rather than a 25% reduction in ocean ventilation or the increase in DOC (Fig. 9).  This dominant control by remineralization of organic matter in comparison to ventilation changes may be link to changes in upwelling and export production.

The increased atmospheric $CO_2$ with radiative forcing simulations of this study agree with other studies of model-simulated change and observed change in the extent of OMZs (Whitney et al. 2007, Karstensen et al. 2008, Stramma et al. 2008, Shaffer et al. 2009, Falkowski et al. 2011). However, the simulations presented here have a greater overall decrease in global oxygen concentration of 9.1% after 300 years of integration for a doubling of $pCO_2$ than previous studies, which range from 1-7% for various $pCO_2$ emissions and integration times (Matear et al. 2000, Bopp et al. 2002, Oschlies et al. 2008, Schmittner et al. 2008, Bopp et al. 2013). The rapid decrease in global dissolved $O_2$ concentration is due to the rapid change in global ocean temperature linked to the 1% business as usual atmospheric $CO_2$ emissions. However, the dissolved oxygen concentrations in the OMZ areas decrease more slowly in the model simulations as compared to the observed trends from Stramma et al., (2008). The study of Stramma et al., (2008) suggests a temperature increase of 0.005 °C $yr^{-1}$ in the Atlantic and Indian Oceans and a temperature decrease by 0.005 °C $yr^{-1}$ for the Pacific Ocean since the 1960s. Most of the expansion of suboxic area in this model study occurs during the first 2000 years of the 30,000-year simulation due to the slow response time, particularly in the deep Pacific Ocean. The atmospheric $pCO_2$ is stabilized at the elevated

$CO_2$ concentrations in the carbon perturbation simulations in this study; therefore, no
recovery is simulated.
In all carbon perturbation simulations the upper boundary of the OMZ cores are
shallower compared to the reference simulation. The shallowest OMZ core is found in the
Indian Ocean OMZ at ~75 meters. Note that the upper boundary of the OMZ is located at 75
m depth because above this depth water masses are influenced by the air sea gas exchange of
the uppermost model layer. The core is not expected to shoal beyond 50 m depth in the
simulations due to the assumption that the atmosphere is at equilibrium with the surface of the
ocean which is simulated as the top 50 meters. The OMZ core of the North Pacific Ocean has
the deepest upper boundary, shoaling approximately 100 meters for the highest $pCO_2$ carbon
perturbation scenario. Downward expansion of the OMZ core is limited by the lower
boundary of the activity-ventilated zone at approximately 2000 meters in the Pacific Ocean.
This depth coincides with the depth of the wind-driven circulation, which remains unchanged
in each simulation, because the same wind stress forcing is applied to all simulations.
Deepening of the eastern South Atlantic OMZ and the Indian Ocean OMZ are also limited to
the bottom boundary of the well-ventilated mixed layer (~1500 meter for the Atlantic and
~1000 meters for the Arabian Sea). The ventilation depth of the Arabian Sea may be
overestimated in the model due to the lack of monsoon variation, which can cause the mixed
layer depth to vary greatly in the Arabian Sea.
The simulated OMZs in the Indian and Atlantic Ocean respond to changes in the temperature-
dependent export production of POC and to changes in dissolved organic carbon. $P_{POC}$
increased in the Indian and Atlantic Ocean in the 2 X and 4 X $CO_2$ simulations and started to
decrease in the 6 X and 8 X $CO_2$ experiments (Fig. 2); however, dissolved organic carbon
increases at higher $pCO_2$ concentrations.  The decrease in $P_{POC}$ may be due to the trapping of
nutrients in the equatorial Pacific Ocean which exhibited a large increase in $PO_4$ at 6 X and 8
X $CO_2$. The limited expansion of the OMZ in the Indian and Atlantic Ocean in the 6 X and 8
X $CO_2$ simulations are the result of oxygen loss due to changes in solubility and to a lesser
degree increased dissolved organic carbon, which increases the amount of oxygen available
for remineralization by the model (Fig. 10). Therefore, oxygen is still consumed in the OMZs
of the Indian Ocean and Atlantic Ocean despite the loss of $P_{POC}$ due to the high amount of
dissolved organic carbon.  Figure 10 shows an increase in mineralization in the Indian and
Atlantic Ocean due to high concentration of DOC regardless of the loss in $P_{POC}$. Furthermore,

the extent of the OMZs in the Indian and Atlantic Oceans appear to be responding to changes in the export of organic matter in response to radiative forcing as well as changes in DOC in simulations of less than 6 times of the preindustrial $pCO_2$ (Fig 11). The extent of the present day OMZ in the Atlantic Ocean has a much higher dissolved oxygen concentration due to cooler water masses than in the northern Indian Ocean. However, the higher salinity of the Atlantic Ocean could lead to greater loss of $O_2$ solubility at higher seawater temperatures as compared to the Indian Ocean or eastern tropical Pacific Ocean for each $pCO_2$ simulation.

The change in the extent of the OMZ in the Pacific Ocean is driven by the change in productivity and export production of POC and increases in remineralization (Fig. 10 and 11). The response of the model to changes in $P_{POC}$ in the Pacific Ocean is stronger than to a reduction in ventilation by 25% (Fig. 9). The increase of export production of POC in the eastern equatorial Pacific OMZ leads to significant horizontal expansion, which is not simulated in the eastern South Atlantic or the Indian Ocean OMZs. The model does not indicate a more significant increase in export production of POC in the cold tongue of the Pacific Ocean as compared to the warm pool in the western Pacific Ocean. However, it is important to note that the simulated $CO_2$-induced seawater temperature change is uniform and therefore the eastern Pacific seawater temperature remains cooler relative to other regions of the Pacific Ocean. The Pacific Ocean OMZ does not shoal as significantly as the Indian Ocean or eastern South Atlantic OMZs but expands horizontally under the area of high productivity. Oxygen loss due to remineralization of organic matter is potentially the main mechanism for simulated expansion of the OMZ in the tropical Pacific Ocean. Figures 10 and 11 include cross sections of the amount of oxygen consumed by the remineralization of organic matter indicating the large influence of organic matter export in the eastern tropical Pacific OMZ as opposed to eastern South Atlantic OMZ.

In the carbon cycle perturbation simulations, the LOZ that currently exists in the western tropical Pacific meets the criteria of a permanent non-seasonal OMZ at approximately 3 X $CO_2$; however, in <2000 yrs a much stronger OMZ core develops in the 4 X $CO_2$ simulation (Fig. 8). The formation occurs northwest of the Gulf of Carpentaria and expands into the Banda Sea and south along the west coast of Australia. The western tropical Pacific OMZ forms in the warm water masses of the Indonesian throughflow (ITF), which brings warm water westward from the Pacific into the Indian Ocean. The OMZ is then expanded by the oxygen-depleted water masses originating from the Leeuwin Current, which flows south

around the west coast of Australia. There is a net loss of export production of POC and a slight increase in DOC in the area suggesting the main control of OMZ core formation in the model is similar to that of the Indian and Atlantic Ocean OMZ expansion. This region is an area of high heat transport between the Pacific and Indian Oceans. The formation of an OMZ could be expected in this area of higher SST; however, it is important to note that the model simulation does not include changes in the intense tidal induced mixing that may affect sea surface temperatures and dissolved oxygen concentrations within the Indonesian throughflow nor any global changes to ocean ventilation. Furthermore, the DO concentration in the OMZ core of the western tropical Pacific Ocean is increased by 15 $\mu$mol kg$^{-1}$ $O_2$ with a 50% reduction in ventilation and therefore the OMZ simulated would not reach the OMZ criteria proposed here at 4 X $CO_2$ and a 50% reduction in ventilation.

## 6 Conclusions

Increased sea surface temperature as a result of $CO_2$ radiative forcing will likely cause expansion of present-day tropical OMZs as well as the possibility of the formation of new oxygen depleted regions. Understanding the extent and the mechanisms for these OMZ expansions and how models respond to changes in expansion mechanism is of the utmost importance in order to more accurately predict environmental changes in these regions. Simulated expansion of the oxygen minimum zones in this model study are greatest in the eastern tropical Pacific Ocean, indicating that the model is sensitivity to the change in export of particulate organic carbon which is overestimated by the model. Total production increases most in the equatorial Pacific leading to the rapid horizontal expansion of the OMZ core. Furthermore,, a change in the ecosystem structure could alter the C:N stoichiometry (carbon overconsumption) and therefore the expansion of the OMZ in the eastern equatorial Pacific Ocean could be reduced due to decrease in the export production of POC.

A rise in $P_{POC}$ (2 X and 4 X $CO_2$ simulations) and dissolved organic carbon (6 X and 8 X $CO_2$ simulations) in conjunction with changes in solubility in the Atlantic Ocean leads to the greatest loss of simulated dissolved oxygen in the intermediate water masses of any of the OMZs. Dissolved oxygen loss causes a greater shoaling and deepening in the eastern tropical South Atlantic OMZ in the model rather than horizontal expansion. The Indian Ocean OMZ is restricted in horizontal expansion; therefore, simulated changes in this OMZ are mostly a vertical expansion of the core, which expands at a similar rate as the eastern tropical South

Atlantic OMZ. This simulated expansion is due loss of solubility and an increase in oxygen available for remineralization due to increased concentration of dissolved organic carbon in a region which is already at very low dissolved oxygen concentrations.

In conclusion, as sea surface temperature increases as a result of $CO_2$ emission the OMZs will expand and strengthen as a result of changes in export of POC, DO, solubility and ventilation. These changes will limit migration and habitat zones resulting in fundamental changes in the marine ecosystem. The loss of dissolved oxygen will also result in changes to the carbon and nitrogen cycles. Any expansion of hypoxia into the photic zone could be detrimental to marine ecosystems. Further research on the expansion of OMZ should include changes in ocean circulation due to changes in density and increased stratification in a comprehensive earth system model (see e.g. Moore et al. 2013). Changes in the ventilation of the ocean waters could lead to changes in both the intensity of the oxygen minimum zones as well as any future expansion.

**Acknowledgements**

We acknowledge all those who have worked on and with the HAMOCC model, most notably the late Dr. Ernst Maier-Reimer and Dr. Virginia Palastanga. Plotting was accomplished on NCAR computers, which are supported by the National Science Foundation. This research is supported by NSF grants EAR-0628336 and OCE-1536630 as well as NSF STEM support and UTA graduate dissertation fellowship.

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

Table 1. List of initial conditions.
Table 2. List of model scenarios.
Figure 1. Atmospheric $pCO_2$ and sea surface temperature increase from the reference run (a)
2 X $CO_2$ (b) 4 X $CO_2$ (c) 6 X $CO_2$ (d) 8 X $CO_2$ for the first 500 years of a 30k year
simulation. The red dashed line indicates the preindustrial $pCO_2$ level.
Figure 2. The difference in particulate organic carbon ($\mu mol\ kg^{-1}$) for (a) 4 X $CO_2$ and the
reference simulation, (b) 6 X $CO_2$ and the reference simulation and (c) 8 X $CO_2$ simulation
and the reference simulation.
Figure 3. Locations of the OMZ at 450 meters depth simulated by HAMOCC 2.0 (reference
experiment) [1] Eastern North Pacific OMZ [2] Eastern South Pacific OMZ [3] Eastern South
Atlantic OMZ [4] Indian Ocean.
Figure 4. Reduced ventilation simulations at (a) reference (100% ventilation), (b) 25%
reduction, (c) 50% reduction, (d) 75% reduction and (e) 100% reduction in ventilation.
Figure 5. Dissolved $O_2$ concentration simulated by (a) the 4 X $CO_2$ experiment without $CO_2$
radiative forcing minus the reference experiment (b) the 4 X $CO_2$ with $CO_2$ radiative forcing
simulation minus reference experiment.
Figure 6. The horizontal expansion of OMZs at 450 meters depth for the Pacific, Atlantic and
Indian Oceans in the (a) 2 X $CO_2$ simulation, (b) 4 X $CO_2$ simulation, (c) 6 X $CO_2$ simulation
and 8 X $CO_2$ simulation.
Figure 7. Simulated vertical distribution of dissolved $O_2$ through the OMZ cores for a)
Eastern North Pacific OMZ [110°W, 10°N], b) Eastern South Pacific OMZ [85°W, 10°S], c)
Eastern South Atlantic OMZ [5°W, 10°S], and d) Indian Ocean OMZ [Gulf of Bengal; 85°E,
7°N] for the 1 X, 4 X and 8 X $CO_2$ simulations (top). The bottom row are finer scale
dissolved oxygen profiles for the OMZ cores e) Eastern North Pacific OMZ, f) Eastern South
Pacific OMZ, g) Eastern South Atlantic OMZ, and h) Indian Ocean OMZ for the 1 X, 4 X and
8 X $CO_2$ simulations. Observations are the annual statistical mean for dissolved oxygen from
the World Ocean Atlas, 2013 (Garcia et al., 2014). Standard error of the mean; upper ocean:
0.54-2.86 $\mu mol\ L^{-1}$, twilight zone: 0.42-2.32 $\mu mol\ L^{-1}$, deep ocean: 0.36-1.98 $\mu mol\ L^{-1}$.
Figure 8. Zonal cross-section at 1.25° N of the formation of the western tropical Pacific OMZ
for the (a) 2 X, (b) 4 X and (c) 8 X $CO_2$ simulations. The OMZ core is located between 130°
E and 150°E.
Figure 9. The difference in the dissolved oxygen concentration between (a) the 50% reduction
in ventilation and the 4 X $CO_2$ simulation with radiative forcing and (b) the 75% reduction in
ventilation and the 4 X $CO_2$ simulation with radiative forcing.
Figure 10. Mechanisms for oxygen loss in the OMZs at 8 X $CO_2$. (a) Reference simulation.
(b) The difference in DO concentrations between 8 X $CO_2$ and the reference simulation. (c)
The difference in DO lost due to changes in solubility between 8 X $CO_2$ and the reference
simulation. (d) The increase in oxygen consumption due to remineralization of organic carbon
between the 8 X $CO_2$ and reference simulation.
Figure 11. Mechanisms for oxygen loss in the OMZs at 4 X $CO_2$. (a) Reference simulation.
(b) The difference in DO concentrations between 8 X $CO_2$ and the reference simulation. (c)
The ifference in DO lost due to changes in solubility between 8 X $CO_2$ and the reference
simulation. (d) The increase in oxygen consumption due to remineralization of organic carbon
between the 4 X $CO_2$ and reference simulation.

1	Table 1. List of initial conditions.

| Water Column | | Atmosphere | |
|---|---|---|---|
| Parameter | Value (mol L$^{-1}$) | Parameter | Value (ppmv) |
| DIC[12] | 2.25 E$^{-3}$ | $CO_2$ | 279.78 |
| Alkalinity | 2.33 (eq) | $O_2$ | 209761 |
| PO$_4$ | 2.54 E$^{-4}$ | | |
| O$_2$ | 1.65 E$^{-4}$ | | |
| Fe dust | 6.0 E$^{-10}$ | | |

Table 2. List of model scenarios.

| Increased $pCO_2$ without Radiative Forcing | Increased $pCO_2$ with Radiative Forcing | Atmospheric $CO_2$ Concentration (ppmv) | Integration Time (years) | Brief Description of the Simulation |
|---|---|---|---|---|
| | | | | |
| **CO$_2$ Stabilization Simulations** | | | | |
| 1 X CO$_2$ | | 279.78 | 30,000 | Reference simulation with preindustrial atmospheric CO$_2$ levels. |
| 2 X CO$_2$_nf | 2 X CO$_2$_f | 559.56 | 30,000 | Experiments with no feedbacks (nf) have an increase of pCO$_2$ of |
| 3 X CO$_2$_nf | 3 X CO$_2$_f | 839.34 | 30,000 | 1% per year without temperature feedbacks. Temperature |
| 4 X CO$_2$_nf | 4 X CO$_2$_f | 1,119.12 | 30,000 | changes are applied in experiments with feedbacks (f) as a |
| 6 X CO$_2$_nf | 6 X CO$_2$_f | 1,678.68 | 30,000 | function of pCO$_2$ after Hansen et al. (1988) resulting in a |
| 8 X CO$_2$_nf | 8 X CO$_2$_f | 2,238.24 | 30,000 | seawater temperature change of 2.8°C, 5.9°C, 8.7°C and 11.5°C for 2 X, 4 X, 6 X and 8 X CO$_2$, respectively. |
| **Kill-Biology Simulations** | | | | |
| **Kill_All_Prod** | | 279.78 | 1000 | Kill_All_Prod is simulated as an extinction simulation with primary productivity (POC, Si, CaCO$_3$) reduced to $1 \times 10^{-20}$ PgC yr$^{-1}$ and present day atmospheric O$_2$ concentrations. |
| **Preindustrial P$_{POC}$ with Increasing Atmospheric pCO$_2$ Simulations** | | | | |
| | 2 X CO$_2$_POC | 559.56 | 10,000 | Experiments for static POC with changing atmospheric pCO$_2$ |
| | 4 X CO$_2$_POC | 1,119.12 | 10,000 | concentrations involve prescribing POC to a preindustrial value |
| | 8 X CO$_2$_POC | 2,238.24 | 10,000 | that does not evolve with model integration. |
| **Reduced Ventilation Simulations** | | | | |
| **Vent_25** | | 279.78 | 10,000 | Experiments with reduction in ventilation include a simulation in which ventilation (vertical, horizontal and meridianal) is reduced by 25%, 50%, 75% and 100%. Atmospheric pCO$_2$ remains at preindustrial concentrations for all experiments. |

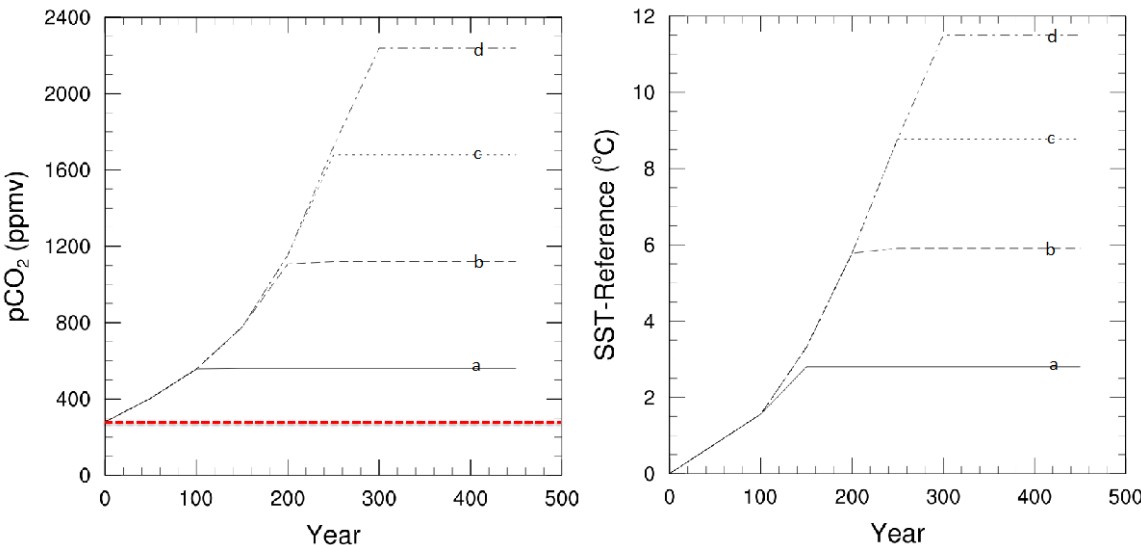

Figure 1. Atmospheric $pCO_2$ and sea surface temperature increase from the reference run

(a) 2 X $CO_2$ (b) 4 X $CO_2$ (c) 6 X $CO_2$ (d) 8 X $CO_2$ for the first 500 years of a 30k year

simulation. The red dashed line indicates the preindustrial $pCO_2$ level.

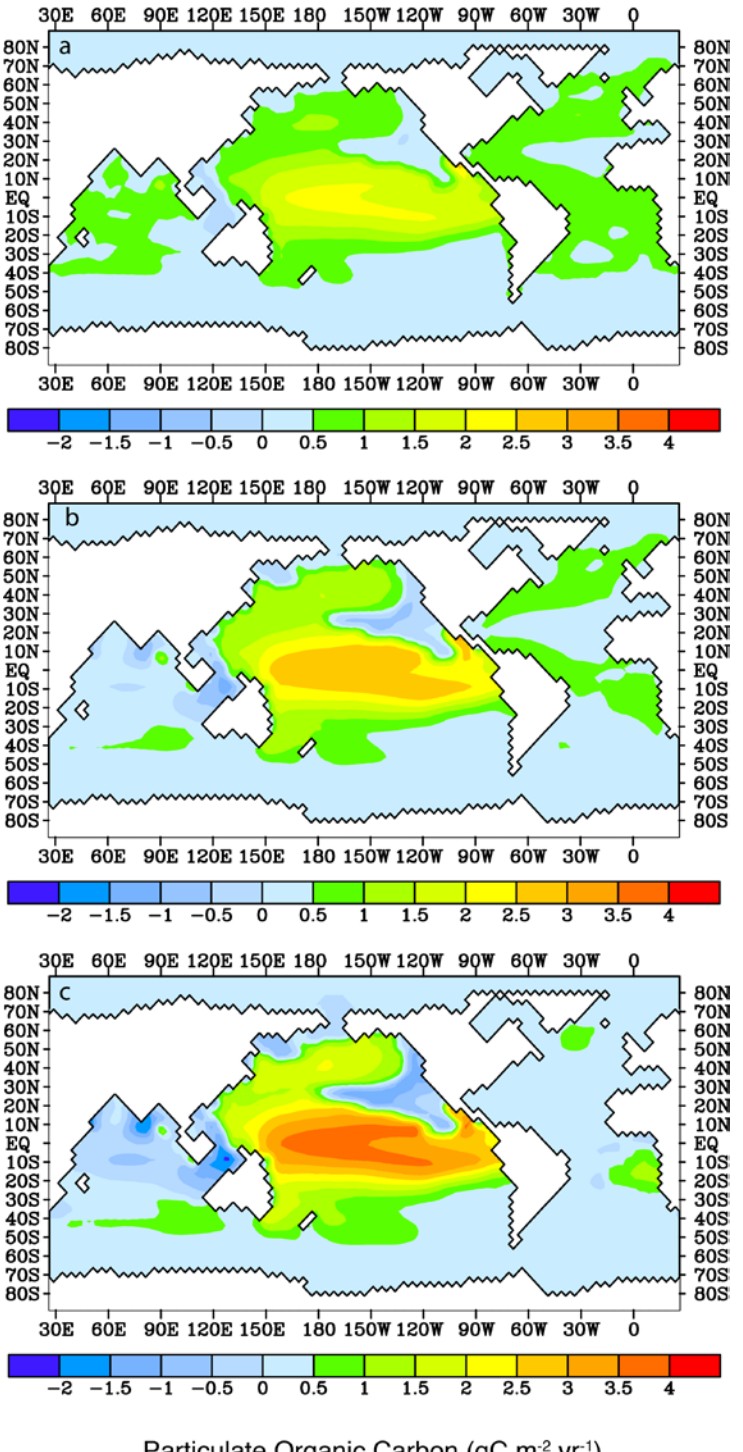

Particulate Organic Carbon (gC m$^{-2}$ yr$^{-1}$)

Figure 2. The difference in particulate organic carbon ($\mu$mol kg$^{-1}$) for (a) 4 X CO$_2$ and the reference simulation, (b) 6 X CO$_2$ and the reference simulation and (c) 8 X CO$_2$ simulation and the reference simulation.

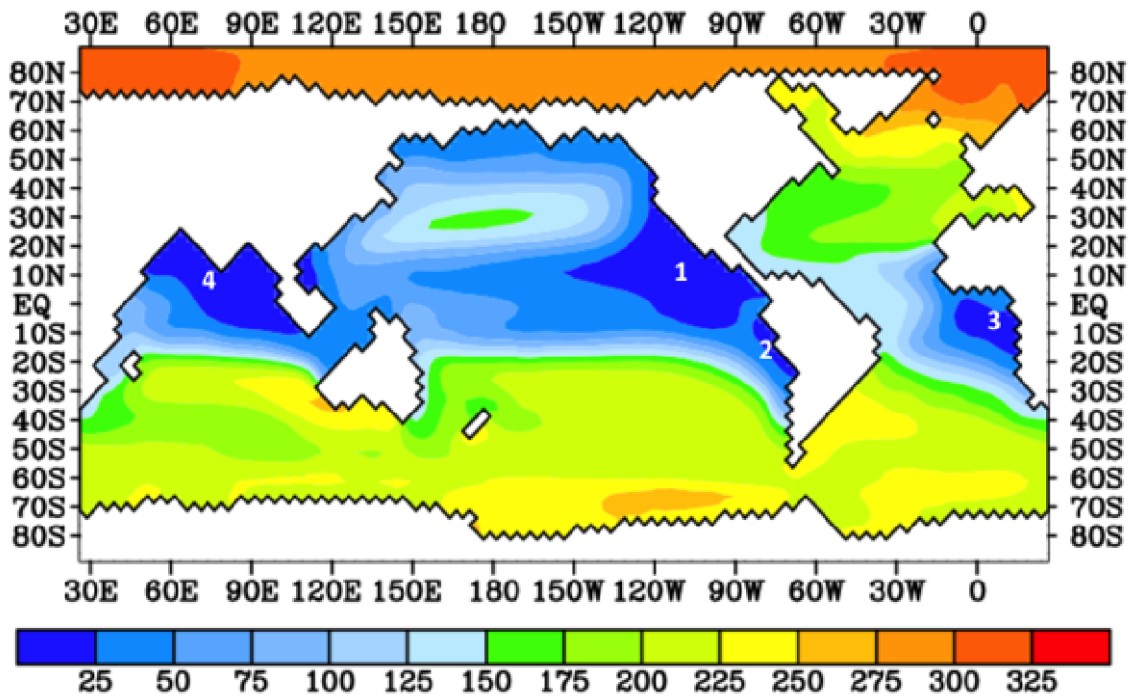

Figure 3. Locations of the OMZ at 450 meters depth simulated by HAMOCC 2.0 (reference experiment) [1] Eastern North Pacific OMZ [2] Eastern South Pacific OMZ [3] Eastern South Atlantic OMZ [4] Indian Ocean.

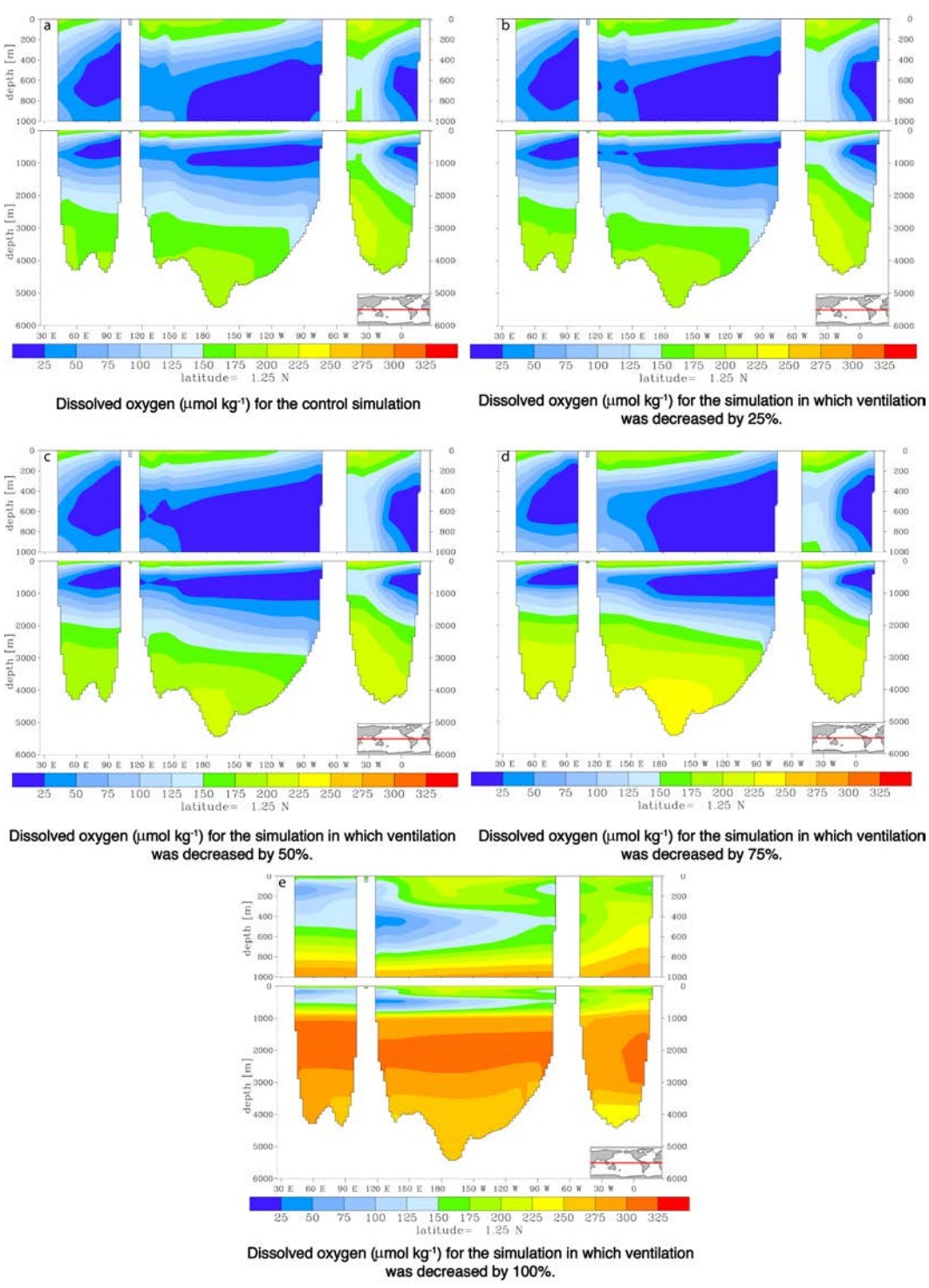

Figure 4. Reduced ventilation simulations at (a) reference (100% ventilation), (b) 25% reduction, (c) 50% reduction, (d) 75% reduction and (e) 100% reduction in ventilation.

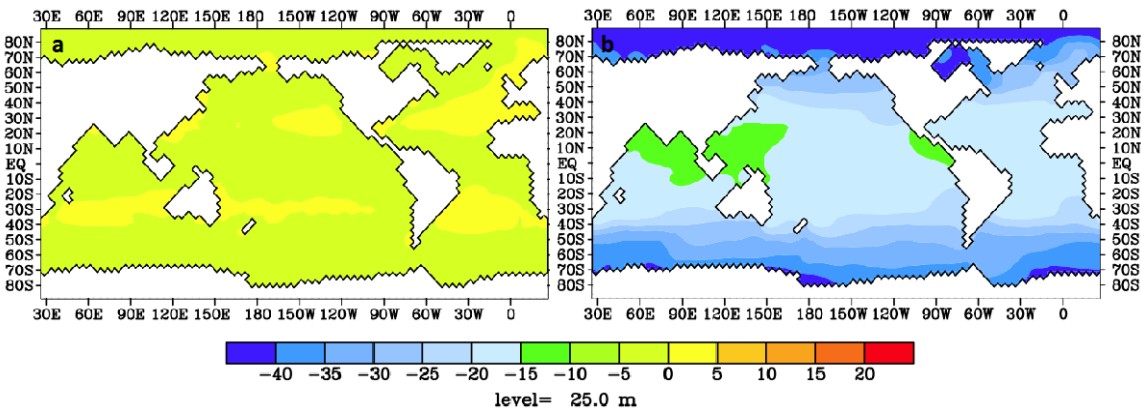

Figure 5. Dissolved $O_2$ concentration simulated by (a) the 4 X $CO_2$ experiment without $CO_2$ radiative forcing minus the reference experiment (b) the 4 X $CO_2$ with $CO_2$ radiative forcing simulation minus reference experiment.

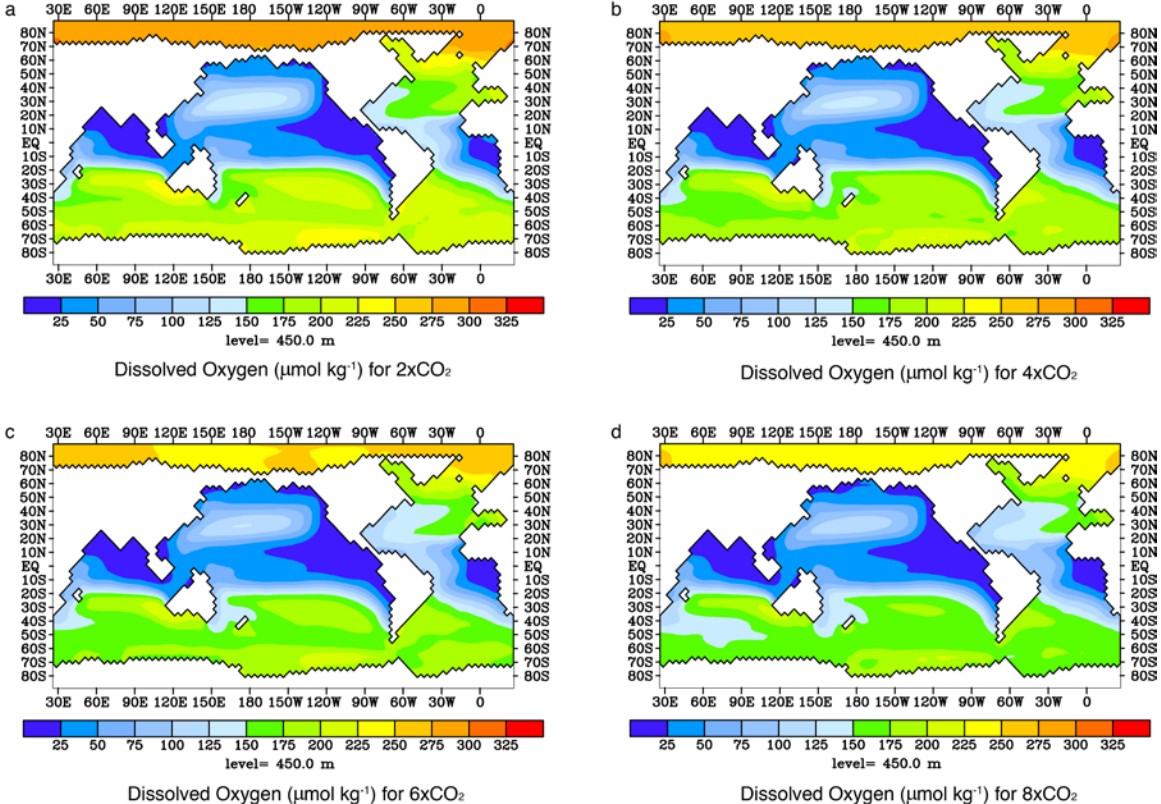

Figure 6. The horizontal expansion of OMZs at 450 meters depth for the Pacific, Atlantic and Indian Oceans in the (a) 2 X $CO_2$ simulation, (b) 4 X $CO_2$ simulation, (c) 6 X $CO_2$ simulation and 8 X $CO_2$ simulation.

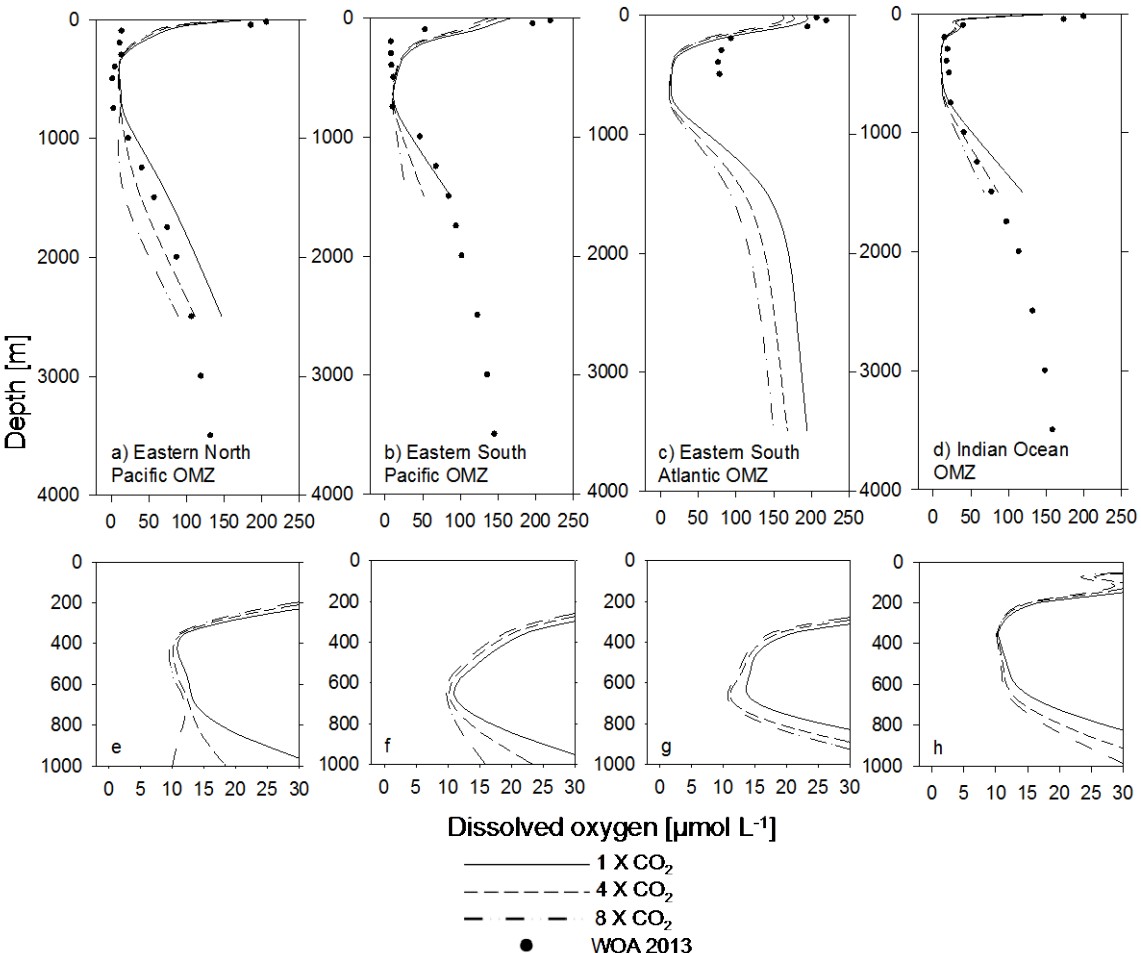

Figure 7. Simulated vertical distribution of dissolved $O_2$ through the OMZ cores for a) Eastern North Pacific OMZ [110°W, 10°N], b) Eastern South Pacific OMZ [85°W, 10°S], c) Eastern South Atlantic OMZ [5°W, 10°S], and d) Indian Ocean OMZ [Gulf of Bengal; 85°E, 7°N] for the 1 X, 4 X and 8 X $CO_2$ simulations (top). The bottom row are finer scale dissolved oxygen profiles for the OMZ cores e) Eastern North Pacific OMZ, f) Eastern South Pacific OMZ, g) Eastern South Atlantic OMZ, and h) Indian Ocean OMZ for the 1 X, 4 X and 8 X $CO_2$ simulations. Observations are the annual statistical mean for dissolved oxygen from the World Ocean Atlas, 2013 (Garcia et al., 2014). Standard error of the mean; upper ocean: 0.54-2.86 μmol $L^{-1}$, twilight zone: 0.42-2.32 μmol $L^{-1}$, deep ocean: 0.36-1.98 μmol $L^{-1}$.

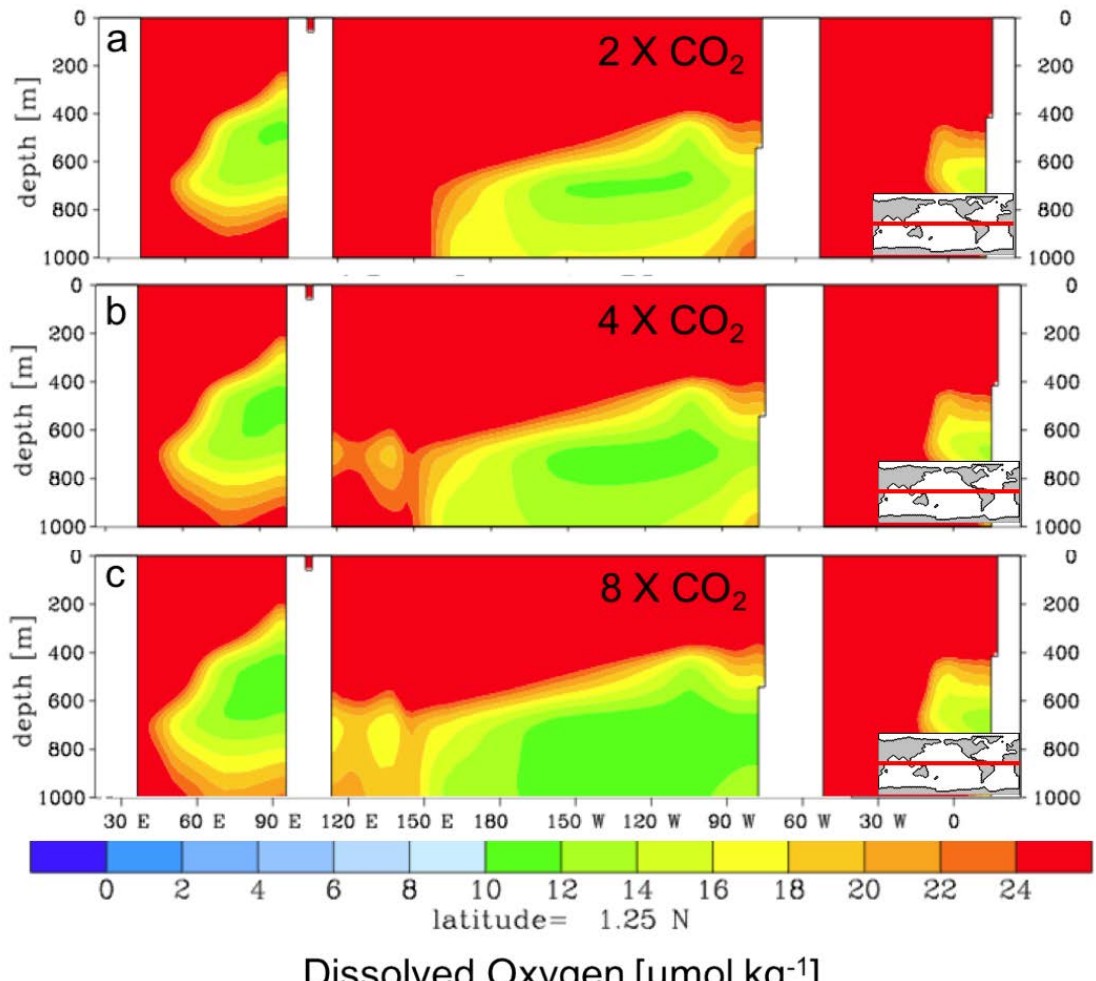

Figure 8. Zonal cross-section at 1.25° N of the formation of the western tropical Pacific OMZ for the (a) 2 X, (b) 4 X and (c) 8 X $CO_2$ simulations. The OMZ core is located between 130°E and 150°E.

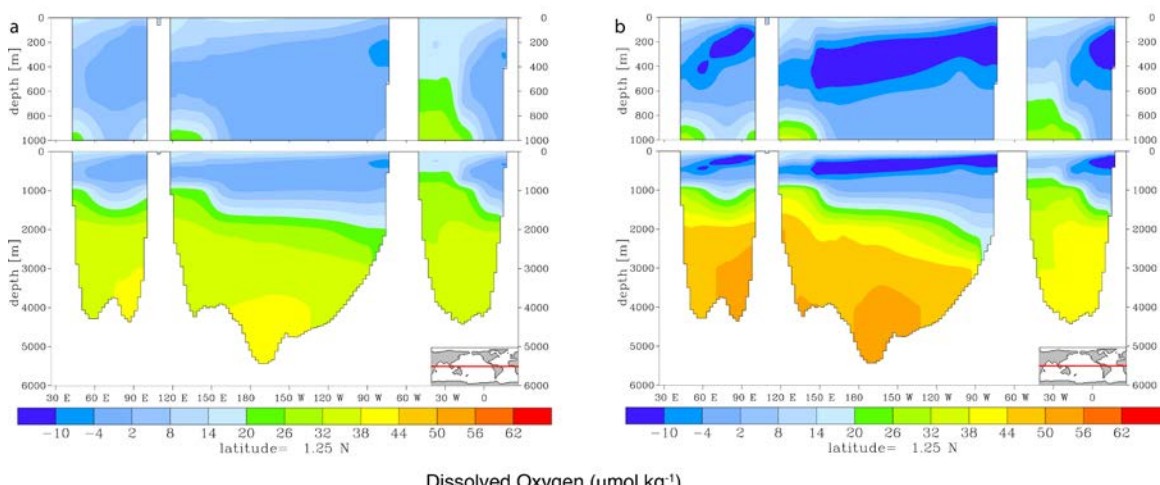

Figure 9. The difference in the dissolved oxygen concentration between (a) the 25% reduction in ventilation and the 4 X CO$_2$ simulation with radiative forcing and (b) the 50% reduction in ventilation and the 4 X CO$_2$ simulation with radiative forcing.

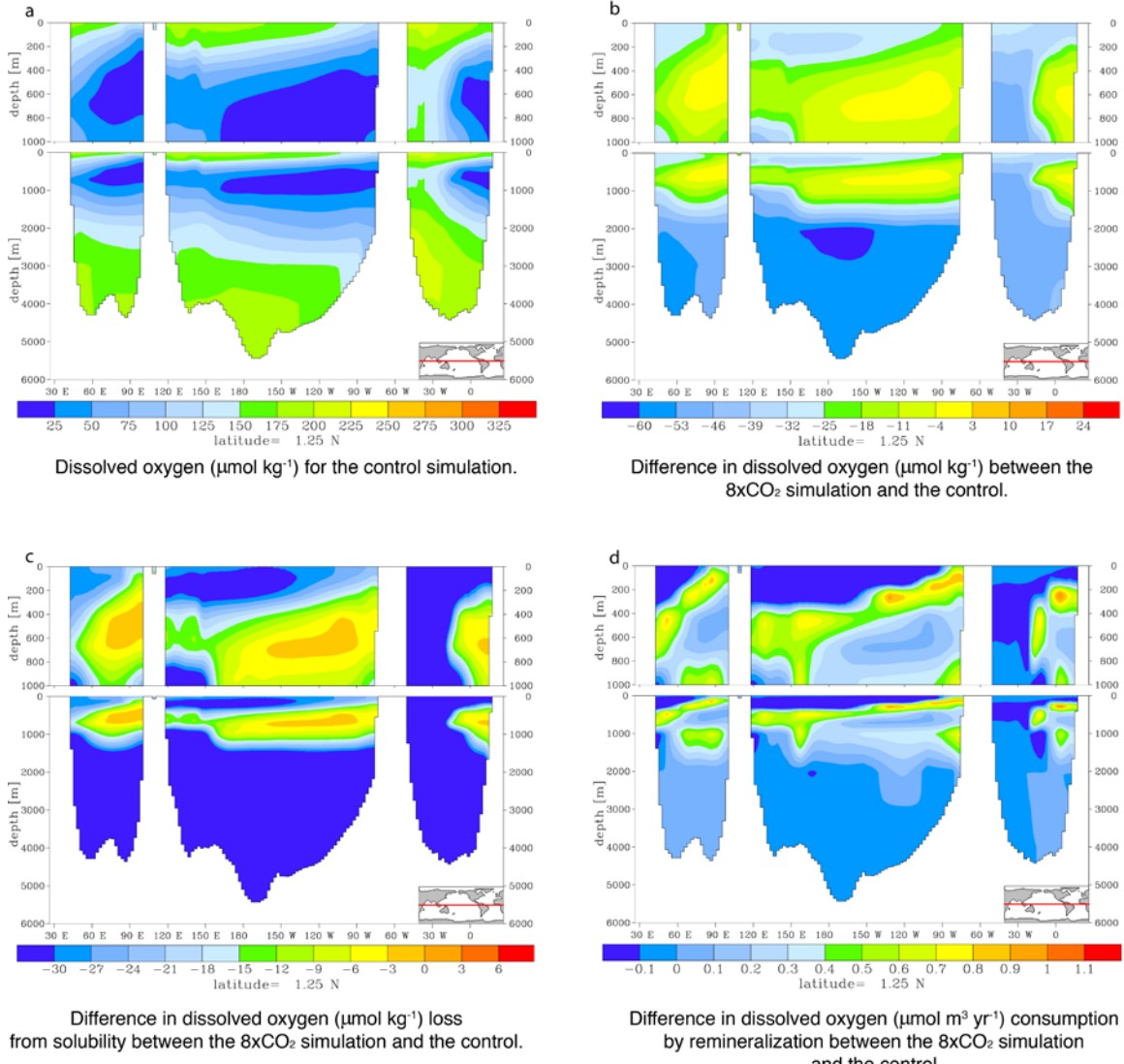

Figure 10. Mechanisms for oxygen loss in the OMZs at 8 X $CO_2$. (a) Reference simulation. (b) The difference in DO concentrations between 8 X $CO_2$ and the reference simulation. (c) The difference in DO lost due to changes in solubility between 8 X $CO_2$ and the reference simulation. (d) The increase in oxygen consumption due to remineralization of organic carbon between the 8 X $CO_2$ and reference simulation.

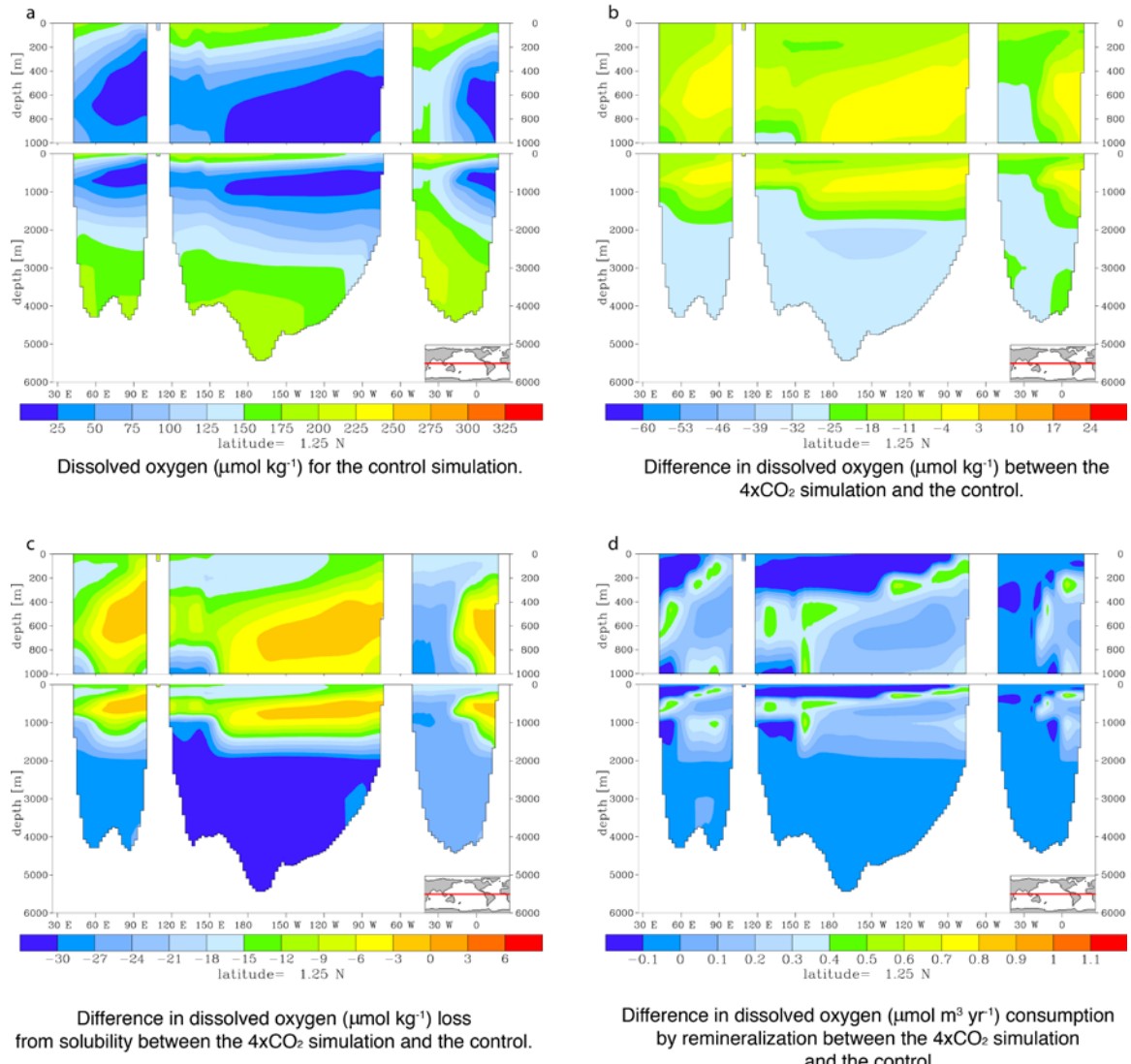

Figure 11. Mechanisms for oxygen loss in the OMZs at 4 X $CO_2$. (a) Reference simulation. (b) The difference in DO concentrations between 4 X $CO_2$ and the reference simulation. (c) The ifference in DO lost due to changes in solubility between 4 X $CO_2$ and the reference simulation. (d) The increase in oxygen consumption due to remineralization of organic carbon between the 4 X $CO_2$ and reference simulation.