# Peer review of "Response of Export Production and Dissolved Oxygen Concentrations in Oxygen Minimum Zones to pCO$_2$ and Temperature Stabilization Scenarios in the Biogeochemical Model HAMOCC 2.0"

_Biogeosciences, 2015_

## Referee Comment (RC1) · Anonymous Referee #1 · 15 Apr 2016

GENERAL ASSESMENT

This manuscript presents the results of a modelling exercise to forecast the global distribution of dissolved oxygen (DO) in the oceans under different CO2 radiative forcing scenarios. It focuses specifically on the expected expansion of the currently existing and forthcoming oxygen minimum zones (OMZ). I have contrasting feelings with this work. On the one side, despite I'm not a modeller, I can appreciate the broad scientific interest of the study and the modelling and interpretation effort done by the authors. On the other side, the fact that the model does not forecast the changes in formation rates of the water masses and, therefore, the ventilation of the ocean interior, is a major weakness of the study. I know that the authors are aware of this limitation and

briefly discuss their implications but, to my understanding, they have omitted the most important process driving the extension of the OMZs.

DETAILLED COMMENTS

Page 1, line 18. As far as I understood, the model does not evaluate only the response to changes in $CO_2$ radiative forcing but to $CO_2$ dissolution in the surface ocean too.

Page 2, lines 2 & 3. Unit of dissolved oxygen concentration are incorrect. They should be $\mu$mol L–1 to be consistent with the units reported in the manuscript. Furthermore, when dissolved oxygen concentrations are reported in $\mu$mol L–1 this means that two samples with the same amount of dissolved oxygen but different temperature will have a different concentration of dissolved oxygen unless you are reporting the concentrations at a fixed temperature. Is this the case? If not, it would be better to report the concentrations in $\mu$mol kg–1, as use to be done in oceanography.

Page 2, line 3. Do you refer to ocean area or you mean ocean volume? I think that in this context ocean volume would be more appropriate. This is applicable to all the times that you refer to ocean area throughout the manuscript.

Page 2, lines 18-21. The description of the factors controlling the distribution of dissolved oxygen in the ocean is incomplete. On the one side, apart from the exchange of dissolved oxygen between the sea surface and the atmosphere, which depend on the solubility of this gas in seawater, the formation of central, intermediate and deep water masses is essential to understand the ventilation of the ocean interior. On the other side, only one element of the organic carbon biological pump is considered: the downward flux of particulate organic carbon. At least, the contribution of dissolved organic carbon (which represents 20% of the organic carbon biological pump) should be mentioned.

Page 2, lines 23–26. It is not just solubility and variations in the biological pump, but ventilation of the ocean interior, i.e. water mass formation rates.

[Figure]

Page 4, lines 26-28. As indicated above, this is only part of the story of the organic matter biological pump. Dissolved and suspended (slow sinking) organic matter is missed despite they can represent about 20% of the downward flux.

Page 5, 16-19. How the increase of temperature of the deep ocean in response to CO2 radiative forcing is modelled without affecting the water mass formation rates and the ventilation time of the ocean interior?

Page 6, lines 7-9. You mean temperature change of the SURFACE ocean, isn't it?

Page 8, line 22 - page 9, line 5. No figure or table is shown; therefore, the reader has just to believe what the authors describe.

Page 9, lines 15-20. A figure should support this description (as supplementary material for example).

Page 9, lines 23 - page 10, Line 19. A figure should support the description of cases other than 4X (as supplementary material for example).

Page 10, line 22 - page 11, Line 6. A figure should support the description of cases other than 4X (as supplementary material for example).

Page 11, lines 18-22. A supplementary figure should support this description

Page 12, lines 11-15. A supplementary figure should support this description

Page 12, lines 16-18. The sentence is confusing. Figure 6 do not show that such an increase by > 300 $\mu$mol L–1 in the deep sea and by > 200 $\mu$mol $\mu$mol L–1 in the intermediate water masses occur. Maybe this is the maximum increases that you detect, but if you use the symbol ">" it looks like this is the minimum increase.

Page 12, lines 29-30. So, what is the significance of all these effort when one of the major drivers of the distribution of dissolved oxygen in the ocean interior is not taken into account?

Page 13, first paragraph. A few sentences should be written about the effect of no considering dissolved organic matter in the transport of organic matter to the ocean interior.

Table 1. Please, define sCO2.

Figure 2. Dissolved oxygen units are missing in the caption.

Figure 3. Dissolved oxygen units are missing in the caption.

Figure 5. Please, add a map of the transect as in Figure 6 and 8. The caption does not indicate which variable are you showing.

Figure 7. To which scenario does panel (d) refers?

Figure 8. "Lost OF OXYGEN due to. . ."; units are not correct: they should be $\mu$mol/m3/y I guess.

---

## Referee Comment (RC2) · Anonymous Referee #2 · 3 May 2016

General comments:

I'm not sure what insights this study can offer on the mechanisms that will regulate the expansion of OMZ in a future warmer climate. Not taking into account changes in stratification and circulation is already a strong limitation, one acknowledged by the authors but I'm not sure it is an acceptable one. Said so, I think the paper is well written and clearly exposed but I also think that the authors could have tried to turn the limitation of not being able to take into account changes in physical transport into an advantage. This could have been done by better separating the changes induced on the OMZ by temperature-driven decrease in solubility of O2 from those induced by temperature-driven changes in the cycling of organic matter. Such separation is not

really achieved for the following reasons:

1- Shouldn't the purpose of the reduced-biology experiment that of being compared to the CO2-radiative experiment to separate the effects of T-driven solubility changes from those deriving from increasing export production? This is not achieved because for the reduced-biology experiment CO2 is kept at pre-industrial level. Therefore, I struggle to see the purpose of this experiment. If more simulations can be performed I suggest to run an experiment in which export production is kept at preindustrial level while O2 solubility is allowed to respond to increasing temperature derived from increasing CO2 radiative forcing.

2- The increase in export production in the Pacific strikes me as a little curious. Is there so much nutrient left at the surface in the control simulation? Because to increase the export production so much by just increasing temperature there's a need for available nutrients at the surface. Maybe a map showing the difference of surface nutrients between control simulation and WOA13 could help to understand this response. In general, I think the authors need to explore more in depth what is causing this increase in export production. For example: Is remineralization dependent on temperature in the model? If so, it could be that increasing T makes for a shallower remineralization recycling more nutrients above 400m depth. This would also explain why O2 consumption decreases deeper down (Figure 8) and more with higher increase in T. There might be a re-circulation loop of nutrients that get trapped in the equatorial circulation. A plot of changes in PO4 would clarify this.

I think the paper should be re-thought and enriched, if possible, with results from new simulations and a more thorough analysis of the responses of the model.

Specific comments:

1-Throughout the paper references to the figures are missing (or at least very infrequent).

2- It would help to synthesize sections 4.1 and 4.2 with figures showing differences between model and observations on a common grid.

3-Page 9, line 4: what is that does not deepen?

4-Maps of changes of distribution of OMZ with CO2 increasing would probably be more illustrative than profiles as those shown in Figure 4.

5-Section 4.6: title here comes a bit as a surprise, reduced atmospheric concentration of O2 not mentioned before.

6-Page 12, lines 11-13: Is the reduced biological pump experiment run at steady-state? Is CO2 allowed to escape the ocean?

7- Page 134, lines 11-14: Could you elaborate this? Why would stronger upwelling be related to slower shoaling of OMZ?

8-Page 16, lines 21-23: Not clear what you mean here.

9- Put variable and units in colorbars of figures.

---

## Author Comment (AC1) · 6 Jun 2016

General assessment; we appreciate the referee's comments and suggestions. We agree with the referee that consideration of changes in ventilation is also of importance for the distribution of dissolved oxygen and the distribution of OMZs in the oceans. However, we originally did not aim to consider circulation changes and to focus on temperature induced changes only. The reason for this was that oxygen and radio-carbon are not very well correlated in the ocean water column and circulation changes may mask the temperature effect. We will, however, include an experiment on the effect of a change in ocean circulation in our revision. HAMOCC is a biogeochemical model designed to long integration with low computational cost to allow adjustment with sediment geochemistry in the timescale of 100,000 yrs. In contrast, comprehensive Earth System Models can run only in a order of several 1,000 yrs. This study focuses on the long-term trend of the oxygen minimum zone by the changes in the biological and solubility pump in response to global temperature on a longer time sales. Interactive changes between ocean circulation and carbon cycle on the millennial time scale with a similar model has been published elsewhere (e.g. in Mikolajewic et al. 2007, Climate Dynamics, or Winguth et al., 2005 JGR). However, for the revision we will add an additional sensitivity experiment where the ocean circulation is shutdown, as an extreme scenario. This will provide insight into the maximum effect of potential circulation changes on dissolved oxygen distribution. .

Page 1 line 18; Yes, we will clarify this sentence to include the distribution of CO2 dissolution. Page 2 lines 2 and 3; We will change the units to umol kg-1. Page 2 line 3; Sentence will be corrected accordingly. Page 2 line 18; We will add a plot and section on DOC; however, since we are focused on changes in POC it was not added to the submission. Page 2 lines 23-26; This is an offline model and flow fields are an input into the model. Therefore we focus on changes in productivity and solubility in this study. Page 4 lines 26-28; We will add changes in TOC which includes both dissolved and particular carbon and discuss these changes in a separate section. Page 5 lines 16-19; To clarify the sensitivity study, we applied a global uniform temperature change in the ocean in each simulation. Page 6 lines 7-9; see major comments. Global mean temperature anomalies have been applied. Pages 8-12 All Supplementary figures requested could be added. Page 12 lines 16-18; The sentence should be corrected to state that "Dissolved oxygen
 17 increases to >300 $\mu$mol L-1 in the deep-sea and >200 $\mu$mol L-1 in the intermediate water masses".
 Page 12 lines 29-30; the purpose of this study is to analyse the strength of the following two OMZ controls: changes in the solubility and biological pump. Page 13 first paragraph. A figure for total organic carbon can be added to the supplementary data as well as adding to information to the text.

Table 1; sCO2 will be defined within the table Figure 2 and 3; units will be added Figure 5; DO with units will be added as well as the map of the transect Figure 7; We will correct the caption to read sea water temperature at 450 meters in the 8 X CO2 simulation Figure 8; units will be corrected to m3

---

## Author Comment (AC2) · 6 Jun 2016

General comments Thank you for your comments and suggestions. I agree that ventilation is an important variable for determining changes in OMZs (see also our respective response to reviewer #1). Though we aimed at focusing on the temperature and CO2 effects on the ocean oxygen distribution, we will add a respective experiment on ocean circulation changes (see also our response to reviewer #1). We focus on changes in the biological pump and how these changes affect ocean interior oxygen and OMZs. We will provide a better separation of the solubility and biological productivity effects with an additional reduced biology experiment. The model does overestimate productivity at the equator due to nutrient trapping (see Najjar et al., 1992, GBC). Productivity

[Figure]

**BGD**

is temperature dependent; however, remineralization is dependent on a fixed Redfield ratio and oxygen consumption (ie POC concentration). A plot of PO4 will be added for clarification in the revision.

Specific comments 1. We will revisit the text to make sure figures are referenced correctly.

2. Section 4.1 and 4.2 will be merged and comparisons to observed data added to the supplementary data.

3. This sentence will be corrected to state that the bottom boundary of the OMZ does not deepen.

4. Changes in the OMZ distribution will be added to the supplementary data. Figure 4 illustrates the changes to the dissolved oxygen profile through the core of the OMZ for each simulation as well as observed data. Illustrating the profile this way allows the reader to evaluate the depth of the oxycline. Unfortunately, this same detail can not be accomplish with the plotting program that accompanies HAMOCC.

5. An introduction to the atmospheric oxygen simulation will be added to page 6 line 13.

6. All experiments including the reduced biology scenario are run from a near-steady state condition and integrated for 30,000 yrs. CO2 and O2 at actively exchanged between the ocean and the atmosphere.

7. This statement will be clarified. Strong upwelling in the tropical Pacific Ocean could transport high-nutrient and oxygen depleted water masses to the surface; however, in the model these waters are at equilibrium with the atmospheric and therefore the effect of the upwelling on DO concentrations is diminished in the model.

8. The model assumes a constant Redfield ratio following Maier-Reimer, 1993, GBC. However, potential changes in the Redfield ratio due to ecosystem dynamics could result in changes in POC and thus the OMZ (e.g. as resulting in a mesocosm experiment,

see Riebesell et al., 2007, Nature). We will discuss these effects in the revised version.

9. Variables and units will be added to the color bar.

---

## Author Response (AR1)

Response of Export Production and Dissolved Oxygen Concentration in Oxygen Minimum Zones to pCO2 and Temperature Stabilization Scenarios in the Biogeochemical Model HAMOCC 2.0

Authors: T. Beaty, A.M.E. Winguth, T. Hughlett, C. Heinze

**Modification to the manuscript:**

In the substantially revised version, we have added an analysis of seven new simulations in order to address the Editors and Reviewers concerns. Four simulations address how the simulated DO concentrations respond to changes in ventilation in the model.  Three additional simulations were added to address changes in solubility by holding POC production at preindustrial levels while increasing $CO_2$ and seawater temperature.  Each of the simulation were discussed and integrated into the manuscript. Addition figures were added as needed to fully discuss these simulations. Second, dissolved organic carbon is considered in the new analysis giving further insight into the changes in DO concentration in the HAMOCC model. Furthermore, the revised manuscript focuses more on long-term changes in the OMZ.  The HAMOCC model is beneficial to investigate long-term biogeochemical cycles that cannot explored by fully couple climate models due to the high computational expenses of these models. Thus, we hope that we have better conveyed the purpose and validity of the work.

List of Changes to the manuscript
- 4 new ventilation simulations
- 3 additional solubility simulation
- The addition of the role of DOC to the discussion, figures and table
- Addition figures to support the text both within the manuscript and as supplementary data

**Point by point comments for Referee #1:**

General assessment; we appreciate the referee's comments and suggestions.  We agree with the referee that consideration of changes in ventilation is of importance also for the distribution of dissolved oxygen and the distribution of OMZs in the oceans. However, we originally did not aim to consider ventilation changes and to focus on temperature induced changes only. The reason for this was that oxygen and radiocarbon are not very well correlated in the ocean water column and ventilation changes may mask the temperature effect. Furthermore, HAMOCC is a biogeochemical model designed to long integration with low computational cost to allow adjustment with sediment geochemistry in the timescale of 10,000 yrs or more. In contrast, comprehensive Earth System Models can run only in a order of several 1,000 yrs. This study focuses on the long-term trend of the oxygen minimum zone by the changes in the biological and solubility pump in response to global temperature on a longer time sales. Interactive changes between ocean ventilation and carbon cycle on the millennial time scale with a similar model has been published elsewhere (e.g. in Mikolajewic et al. 2007, Climate Dynamics, or Winguth et al., 2005 JGR).   We have, however, included an experiment on the effect of a change in ocean ventilation in our revision. Additionally a sensitivity experiment was simulated where the ocean ventilation is shutdown, as an extreme scenario. This will provide insight into the maximum effect of potential ventilation changes on dissolved oxygen distribution in the biogeochemical model.

Page 1 line 18; Yes, we have clarify this sentence to include the distribution of $CO_2$ dissolution.

Page 2 lines 2 and 3; We have change the units of dissolved oxygen to umol kg-1 throughout the manuscript.

Page 2 line 3; Sentence has been corrected accordingly.

Page 2 line 18; We have add a plots, results and discussions on the response DOC in the model.

Page 2 lines 23-26; This is an offline model and flow fields are an input into the model. Therefore we focus on the long term response of DO to changes in POC export production and changes in oxygen solubility in this study.

Page 4 lines 26-28; We have add plots, results and discussions on the response DOC in the model.

Page 5 lines 16-19; To clarify the sensitivity study, we applied a global uniform temperature change in the ocean in each simulation.

Page 6 lines 7-9; See major comments. Global mean temperature anomalies have been applied.

Pages 8-12 Supplementary figures have been added to support the manuscript.

Page 12 lines 16-18; The sentence has be corrected to state that Dissolved oxygen increases to >300 $\mu mol\ L^{-1}$ in the deep-sea and >200 $\mu mol\ L^{-1}$ in the intermediate water masses.

Page 12 lines 29-30; the purpose of this study is to analyze the strength of the following two OMZ controls: changes in the solubility and biological pump.

Page 13 first paragraph. A figure for dissolved organic carbon in conjunction with particulate organic carbon was added to the supplementary data.

Table 1; The 's' in $sCO_2$ will be removed from the table. The 's' is an indicator meaning within the water column in the model and is not necessary in the table.

Figure 2, 5, 7, 8 Many figures have changed in this revision but all corrections were applied if applicable

**Point by point comments for Referee #2:**

General comments

Thank you for your comments and suggestions. I agree that ventilation is an important variable for determining changes in OMZs (see also our respective response to reviewer #1). Though we aimed at focusing on the temperature and $CO_2$ effects on the ocean oxygen distribution, we have added respective experiments on ocean circulation changes (see also our response to reviewer #1). We focus on changes in the biological pump and how these changes affect ocean interior oxygen and OMZs. We have provided a better separation of the solubility and biological productivity effects in additional experiments were POC remains constant and $pCO_2$ is increased. The model does overestimate productivity at the equator due to nutrient trapping (see Najjar et al., 1992, GBC). Productivity is temperature dependent; however, remineralization is dependent on a fixed Redfield ratio and oxygen consumption (i.e. POC concentration). A plot of PO4 has been added to the supplementary data for clarification.

Specific comments
1. We have revisited the text to make sure figures are referenced correctly.
2. Either by merging or separating many sections changed in the manuscript. Also, comparisons to observed data has been added to the supplementary data.
3. This sentence will be corrected to state that the bottom boundary of the OMZ does not deepen.
4. Figure 4 illustrates the changes to the dissolved oxygen profile through the core of the OMZ for each simulation as well as observed data. Illustrating the profile this way allows the reader to evaluate the depth of the oxycline. Unfortunately, this same detail cannot be accomplish with the plotting program that accompanies HAMOCC.
5. The atmospheric oxygen simulation has been removed from the manuscript.

6. All experiments including the reduced biology scenario are run from a near-steady state condition and integrated for 30,000yrs with the exception of the new ventilation simulations and steady POC. These simulations were reduced to 10,000 years since max DO loss occurs ~ 2000 years after max pCO2 is reached. $CO_2$ and $O_2$ are actively exchanged between the ocean and the atmosphere.
7. This statement has been removed from the manuscript. Nonetheless, to clarify strong upwelling in the tropical Pacific Ocean could transport high-nutrient and oxygen depleted water masses to the surface; however, in the model these waters are at equilibrium with the atmospheric and therefore the effect of the upwelling on DO concentrations is diminished in the model.
8. The model assumes a constant Redfield ratio following Maier-Reimer, 1993, GBC. However, potential changes in the Redfield ratio due to ecosystem dynamics could result in changes in POC and thus the OMZ (e.g. as resulting in a mesocosm experiment, see Riebesell et al., 2007, Nature).
9. Variables and units have been added to the color bar.

**Response of Export Production and Dissolved Oxygen Concentrations in Oxygen Minimum Zones to pCO$_2$ and Temperature Stabilization Scenarios in the Biogeochemical Model HAMOCC 2.0**

T. Beaty[1], C. Heinze[2,3], T. Hughlett[4], and A. M. E. Winguth[4],

[1]{New Mexico Consortium-Biological Laboratory, Los Alamos, New Mexico}

[revised manuscript text omitted]

Hasselmann (1987) and Maier-Reimer (1993), and which has been expanded to include an iron cycle, sedimentary phosphorus cycle, and improved atmospheric dust parameterization (Palastanga et al. 2011, Palastanga et al. 2013). HAMOCC was originally developed by

Maier-Reimer and Hasselmann (1987) and Maier-Reimer (1993). The annually averaged version is computationally very economical and suitable for long-term carbon cycle simulations of several 10,000 years (Maier-Reimer and Hasselmann, 1987, Heinze and Maier-

Reimer, 1999). The model utilizes an E-grid (Arakawa and Lamb 1977) and has a horizontal resolution of ~3.5° x 3.5° with grid points 1.25° north and south of the equator to resolve the equatorial upwelling belt. The model contains 11 layers (centered at 25,75,150, 250, 450, 700,

1000, 2000, 3000, 4000, and 5000 meters) with a total depth of 5000 meters (Heinze et al.

1999, Heinze et al. 2006, Heinze et al., 2009). HAMOCC 2.0 includes a sediment module with porewater and solid components that are coupled by a reaction rate. The sediment module includes one 10 cm thick layer of bioturbated sediment, which is further divided into

11 sub-layers. A more detailed description of the sediment module can be found elsewhere (Heinze et al. 1991, Heinze et al. 1999, Heinze 2004).

The annually averaged version is computationally very economical and suitable for long-term carbon cycle simulations of several 10,000 years. Long-term integrations are possible with

HAMOCC because of it coarse temporal and spatial resolution and because of the computational efficient solution tracer equations by an upstream formulations (Maier-Reimer and Hasselmann, 1987, Heinze and Maier-Reimer, 1999) that uses the prescribed annual average circulation and hydrography of the Large Scale Geostrophic (LSG) ocean general circulation model (Maier-Reimer et al., 1993; Winguth et al., 1999).

Transport of tracers is simulated using present-day flow and hydrographic fields (Winguth et al., 1999) from the Hamburg Large-Scale Geostrophic (LSG) model (Maier-Reimer et al.

1993). The advection of tracers is iteratively solved by an upstream formulation (Maier-

Reimer and Heinze 1993). Atmospheric $CO_2$ and $O_2$ are exchanged between the ocean surface (top 50 m) and zonally mixed atmospheric boxes. The air-sea gas exchange of $CO_2$ is
determined by the difference in the partial pressure of $CO_2$ in the sea surface and the
atmospheric $pCO_2$, the gas transfer velocity, and the requirement for a full equilibration of the
surface layer inorganic carbon system. The gas exchange of oxygen is an order of magnitude
faster than that of $CO_2$. Oxygen exchange is carried out according to a fixed transfer velocity
and is assumed to be at equilibrium between the atmospheric layer and the surface water at
the temperature and salinity-dependent saturation level. The solubility of dissolved oxygen
depends on temperature, salinity and pressure (Weiss 1970). The $O_2$ flux into the atmosphere
is neglected since the atmospheric concentration of $O_2$ is by far larger than the DO
concentration at the ocean surface.

The temperature-dependent annual export production of particulate organic carbon (POC) and
opal from the euphotic zone is calculated via Michaelis-–Menten kinetics (Parsons and
Takahashi 1973) and $CaCO_3$ production is dependent on the particulate organic and opal
production. This relationship is based on the assumption that in the present day ocean there is
a dominance of the silicate producers (e.g. diatoms) over the calcareous plankton (e.g.
coccolithophores) (Falkowski et al. 2007). The POC export from the surface into the deep sea
is determined from organic carbon production in the uppermost layer and then transported to
the deep with a uniform sinking rate of 120 m $day^{-1}$. Remineralization of organic matter
depends on the availability of oxygen for consumption in the water column. Remineralization
of POC occurs as long as dissolved $O_2$ is larger than the minimum $O_2$ concentration $[O_{2min}]$ =
$10^{-5}$ mol $L^{-1}$ for bacterial decomposition of POC. A more detailed description of the model
can be found elsewhere (Maier-Reimer and Hasselmann 1987, Heinze et al. 1991, Maier-
Reimer and Heinze 1999, Heinze et al. 1999, Palastanga et al. 2011, Palastanga et al. 2013,
Beaty-Sykes 2014).

**3    Experimental Design**

The annually averaged version of the model was integrated to quasi-equilibrium state (200
kyr) with a stable atmospheric $CO_2$ concentration of 279.78 ppmv. The reference experiment
as well asand all OMZ sensitivity experiments is are started from the near-equilibrium state
and integrated for 30,000 yrs. For the reference experiment, the model is forced withfrom
flowflow fields from the a LSG simulation. The globally averaged potential temperature and

1. salinity are  3.78°C and  of 34.8 psu

2. respectively (Winguth et al. 1999).

3. Carbon cycle sensitivity experiments are conducted in  three sets of scenarios. The first set

4. of scenarios consists of a perturbation of the atmospheric $CO_2$ concentration relative to

5. preindustrial atmospheric levels (p$CO_{2ref}$; PAL) of 2 X $CO_2$, 4 X $CO_2$, 6 X $CO_2$, and 8 X $CO_2$

6. to explore the sensitivity of distribution of dissolved oxygen concentration to rising

7. atmospheric p$CO_2$ level. In these simulations, all other boundary conditions and model

8. parameters are kept at preindustrial levels (Table 1). In a second set of experiments the p$CO_2$

9. levels are accompanied by the associated changes of temperature at the sea surface as well as

10. in the deep sea to investigate the response of the dissolved oxygen distribution to increases in

11. $CO_2$ radiative forcing. In a third set of experiments; POC is kept at preindustrial level to

12. explore the relative strength of loss of $O_2$ solubility and oxygen consumption by

13. remineralization. The preindustrial POC experiments are simulated with at atmospheric $CO_2$

14. concentrations of 2 X, 4 X and 8 X $CO_2$. Stabilization scenarios and brief descriptions are

15. listed in Table 2.

16. In  all $CO_2$ perturbation scenarios atmospheric p$CO_2$ is increased from preindustrial levels

17. by 1% each year (t) until the perturbed atmospheric p$CO_2$ (p$CO_{2pert}$) is stabilized at its

18. maximum level (p$CO_{2max}$) by

19. $$for\ pCO_2 < pCO_{2max}: pCO_{2pert} = pCO_{2ref}(1 + 0.01)^t$$

20. $$and\ for\ pCO_2 \geq pCO_{2max}: pCO_{2pert} = pCO_{2max}. \tag{1}$$

21. The 1% increase of atmospheric $CO_2$ concentration follows the IPCC (2013) business as usual

22. scenario and is stabilized after 70 years for doubling of preindustrial p$CO_2$ (see also Winguth

23. et al. 2005). The second set of carbon perturbation scenarios includes the feedback of

24. increasing seawater temperature due to rising atmospheric p$CO_2$ (Fig. 1). Temperature

25. increases as a function of the 1% increase per time step of atmospheric p$CO_2$ and is

26. determined using Eq. 2 from Hansen et al., (1988) for the radiative forcing of $CO_2$ with the

27. addition of a climate model sensitivity of $A_t$=0.6870.

28. $$\Delta T = A_t\ 6.3 \ln\left(\frac{pCO_2}{pCO_{2ref}}\right) \tag{2}$$

29. Therefore a doubling of p$CO_2$ results in a homogeneous increase in temperature of ~3°C,

30. which is consistent with the estimate of Archer (2005) and Hansen et al. (1988). Note that this enhanced sensitivity includes climate feedbacks whereas the direct $CO_2$ warming for 2 X $CO_2$

is ~1.2°C (Ruddiman 2001, Houghton 2004). The resultant temperature change of the ocean for the doubling of $pCO_2$ for 2 X $CO_2$, 4 X $CO_2$, 6 X $CO_2$, and 8 X $CO_2$ is 2.8°C, 5.9°C,

8.7°C, and 11.5°C respectively (Fig. 1). The temperature change is applied at all depths of the ocean. Solubility and chemical kinetic equilibrium constants of the carbon cycle are adjusted to the changes in $pCO_2$ and temperature at each time step in the temperature feedback experiments.

In addition to experiments with increased $pCO_2$ with and without radiative forcing a reduced biology scenario is added in which primary productivity and export (Si, $CaCO_3$, and organic carbon) is set to zero following the approach of Maier-Reimer et al. (1996). The reduced biology scenario is simulated with preindustrial $pCO_2$ (279 ppmv; Table 2).

In addition to Another scenario withexperiments with increased $pCO_2$ with and without radiative forcing absolutely depleted primary and export production (Si, $CaCO_3$, and organic carbon) and a preindustrial $pCO_2$ values (279 ppmv; Table 2) reduced biology scenario has been carried out.
[revised manuscript text omitted]
 compared to that of the tropical eastern Atlantic and Indian Ocean OMZs may be related to difference in solubility as well as linked to a stronger upwelling in the tropical eastern Pacific

Ocean. Stronger upwelling in the tropical Pacific Ocean could transport high nutrient and oxygen depleted water masses to the surface; however, in the model these water are at equilibrium with atmosphere and therefore the effect of upwelling on DO concentration is diminished in the model. Downward expansion of the OMZ core is limited by the lower boundary of the activity-ventilated zone at approximately 2000 meters in the Pacific Ocean.

[revised manuscript text omitted]

Pacific OMZ forms in the warm water masses of the Indonesian throughflow (ITF), which brings warm water westward from the Pacific into the Indian Ocean. The OMZ is then expanded by the oxygen-depleted water masses originating from the Leeuwin Current, which flows south around the west coast of Australia. The controlling mechanism of the formation of the new OMZ core in the model is similar to that of the Indian Ocean OMZ expansion. There is a net loss of export production of POC and a slight increase in DOC in the area suggesting the main control of OMZ core formation in the model is similar to that of the Indian and Atlantic Ocean OMZ expansion. is loss of $O_2$ solubility due to increased sea surface temperature (SST) in an area of 
[revised manuscript text omitted]
 7. (a) Difference in export production of POC between the 8 X $CO_2$ experiment and reference experiment, (b) dissolved oxygen at 450 m depth for the 8 X $CO_2$ experiment, (c) difference in air-sea gas exchange between the 8 X $CO_2$ experiment and the reference experiment, and (d) sea water temperature at 450 m depth. The numbers indicate the OMZ locations; [1] Eastern tropical North Pacific; 110°W, 10°N. [2] eastern tropical South Pacific; 85°W, 10°S. [3] eastern tropical South Atlantic; 5°W, 10°S. [4] Indian Ocean (Gulf of Bengal); 85°E, 7°N.

Figure 8. (a) Lost due to remineralization of particulate organic carbon for the reference run [μmol $m^{-2}yr^{-1}$]. Difference between the loss of oxygen due to remineralization between (b) 2 X $CO_2$ and reference run, (c) 4 X $CO_2$ and reference run (d) 8 X $CO_2$ and reference experiment. Figure 8. Zonal cross-section at 1.25° N of the formation of the western tropical Pacific OMZ for the (a) 2 X, (b) 4 X and (c) 8 X $CO_2$ simulations. The OMZ core is located between 130° E and 150°E.

[revised manuscript text omitted]

Dissolved oxygen (μmol/kg) for the control simulation

Dissolved oxygen (μmol/kg) for the simulation in which ventilation was decreased by 25%.

Dissolved oxygen (μmol/kg) for the simulation in which ventilation was decreased by 50%.

Dissolved oxygen (μmol/kg) for the simulation in which ventilation was decreased by 75%.

Dissolved oxygen (μmol/kg) for the simulation in which ventilation was decreased by 100%.

[Figure]

Dissolved oxygen μmol L⁻¹

Figure 4. Reduced ventilation simulations at (a) reference (100% ventilation), (b) 25% reduction, (c) 50% reduction, (d) 75% reduction and (e) 100% reduction in ventilation.

[Figure]

Figure 35. Dissolved $O_2$ concentration simulated by (a) the 4 X $CO_2$ experiment without $CO_2$ radiative forcing minus the reference experiment (b) the 4 X $CO_2$ with $CO_2$ radiative forcing simulation minus reference experiment.

[Figure]

Figure 6. The horizontal expansion of OMZs at 450 meters depth for the Pacific, Atlantic and Indian Oceans in the (a) 2 X $CO_2$ simulation, (b) 4 X $CO_2$ simulation, (c) 6 X $CO_2$ simulation and 8 X $CO_2$ simulation.

Commented [TS2]: Figure added at the request of reviewer 1.  Addition plots will be in supplementary data

[Figure]

Figure 47. Simulated vertical distribution of dissolved $O_2$ through the OMZ cores for a) Eastern North Pacific OMZ [110°W, 10°N], b) Eastern South Pacific OMZ [85°W, 10°S], c) Eastern South Atlantic OMZ [5°W, 10°S], and d) Indian Ocean OMZ [Gulf of Bengal; 85°E, 7°N] for the 1 X, 4 X and 8 X $CO_2$ simulations (top). The bottom row are finer scale dissolved oxygen profiles for the OMZ cores e) Eastern North Pacific OMZ, f) Eastern South Pacific OMZ, g) Eastern South Atlantic OMZ, and h) Indian Ocean OMZ for the 1 X, 4 X and 8 X $CO_2$ simulations. Observations are the annual statistical mean for dissolved oxygen from the World Ocean Atlas, 2013 (Garcia et al., 2014). Standard error of the mean; upper ocean: 0.54-2.86 μmol $L^{-1}$, twilight zone: 0.42-2.32 μmol $L^{-1}$, deep ocean: 0.36-1.98 μmol $L^{-1}$.

[Figure]

[Figure]

Figure 58. Zonal cross-section at 1.25° N of the formation of the western tropical Pacific OMZ for the (a) 2 X, (b) 4 X and (c) 8 X CO₂ simulations. The OMZ core is located between 130°E and 150°E.

[Figure]

[Figure]

[Figure]

Figure 9. The difference in the dissolved oxygen concentration between (a) the 25% reduction in ventilation and the 4 X $CO_2$ simulation with radiative forcing and (b) the 50% reduction in ventilation and the 4 X $CO_2$ simulation with radiative forcing.

[Figure]

Figure 10. Mechanisms for oxygen loss in the OMZs at 8 X $CO_2$. (a) Reference simulation. (b) The difference in DO concentrations between 8 X $CO_2$ and the reference simulation. (c) The difference in DO lost due to changes in solubility between 8 X $CO_2$ and the reference simulation. (d) The increase in oxygen consumption due to remineralization of organic carbon between the 8 X $CO_2$ and reference simulation.

[Figure]

a      Dissolved oxygen (μmol/kg) for the control simulation.

b      Difference in dissolved oxygen (μmol/kg) between the 4xCO₂ simulation and the control.

c      Difference in dissolved oxygen (μmol/kg) loss from solubility between the 4xCO₂ simulation and the control.

d      Difference in dissolved oxygen (μmol m³ yr⁻¹) consumption by remineralization between the 4xCO₂ simulation and the control.

[Figure]

Difference in export production of POC between the 8 X $CO_2$ experiment and reference experiment [gC $m^{-2}yr^{-1}$]

Dissolved $O_2$ at 450 meters depth [µmol $L^{-1}$]

Difference in air-sea gas exchange between the 8 X $CO_2$ experiment and the reference experiment [µmol $cm^{-2}yr^{-1}$]

Sea-water temperature at 450 meters depth [°C]

Figure 7. (a) Difference in export production of POC between the 8 X $CO_2$ experiment and reference experiment, (b) dissolved oxygen at 450 m depth for the 8 X $CO_2$ experiment, (c) difference in air-sea gas exchange between the 8 X $CO_2$ experiment and the reference experiment, and (d) sea water temperature at 450 m depth. The numbers indicate the OMZ locations; [1] Eastern tropical North Pacific; 110°W, 10°N. [2] eastern tropical South Pacific; 85°W, 10°S. [3] eastern tropical South Atlantic; 5°W, 10°S. [4] Indian Ocean (Gulf of Bengal); 85°E, 7°N.

[Figure]

Figure 11. Mechanisms for oxygen loss in the OMZs at 4 X CO₂. (a) Reference simulation. (b) The difference in DO concentrations between 4 X CO₂ and the reference simulation. (c) The ifference in DO lost due to changes in solubility between 4 X CO₂ and the reference simulation. (d) The increase in oxygen consumption due to remineralization of organic carbon between the 4 X CO₂ and reference simulation.

Figure 8. (a) Lost due to remineralization of particulate organic carbon for the reference run [μmol m⁻²yr⁻¹]. Difference between the loss of oxygen due to remineralization between (b) 2 X CO₂ and reference run, (c) 4 X CO₂ and reference run (d) 8 X CO₂ and reference experiment.

---

## Author Response (AR2)

Response to reviewer for Response of Export Production and Dissolved Oxygen Concentrations in Oxygen Minimum Zones to pCO2 and Temperature Stabilization Scenarios in the Biogeochemical Model HAMOCC 2.0

Reviewer comment 1:
Page 6, Lines 16-17. This means that the temperature of the whole volume of the ocean, including the deep water of the Arctic and Antarctic origin is going to increase by 2.8, 5.9, 8.7 and 11.5 °C

Author response: The change in average global seawater temperature is the same throughout the ocean. The increases in seawater temperature are an increase in the global average temperature.  Therefore, the models global average seawater temperature is 19.4 °C and at 2 X CO2 the average temperature is 22.2 °C. This does not imply that the ocean temperature is uniform at 22.2 °C only the increase is uniform; therefore, poles stay much cooler than the equatorial region.

Reviewer comment 2:
Page 6, lines 24-27. If u, v and w are reduced by 25% to 100% you cannot say that convection does not change.

Author response: Yes, you are correct.  The convection was not explicitly changed in the model but would be affected t=by the changes in velocities.  The sentence has been corrected to read ' Diffusion is not changed in these experiments and remains at preindustrial reference simulation values.'

Reviewer comment 3:
Page 9, line 12. You mean DO rather than DOC, isn't it?

Author response: No, this line is correct.  As POC decreases in the OMZ area the DOC increases. The simulated OMZ expands even with the sluggish velocities up to the 50% reduction in v, u and w.  We believe this is due to the increase in dissolved organic carbon.  At reduction of greater than 50% the OMZ contract and both POC and DOC are reduced.

Reviewer comment 4:
Page 14, lines 12-13. Please, edit this sentence.

Author response: Thank you for pointing this out. The sentence has been corrected to read '
[revised manuscript text omitted]

Dissolved oxygen (μmol kg⁻¹) for the control simulation.

Difference in dissolved oxygen (μmol kg⁻¹) between the 8xCO₂ simulation and the control.

Difference in dissolved oxygen (μmol kg⁻¹) loss from solubility between the 8xCO₂ simulation and the control.

Difference in dissolved oxygen (μmol m³ yr⁻¹) consumption by remineralization between the 8xCO₂ simulation and the control.

Figure 10. Mechanisms for oxygen loss in the OMZs at 8 X CO₂. (a) Reference simulation. (b) The difference in DO concentrations between 8 X CO₂ and the reference simulation. (c) The difference in DO lost due to changes in solubility between 8 X CO₂ and the reference simulation. (d) The increase in oxygen consumption due to remineralization of organic carbon between the 8 X CO₂ and reference simulation.

[Figure]

Dissolved oxygen (μmol kg⁻¹) for the control simulation.

Difference in dissolved oxygen (μmol kg⁻¹) between the 4xCO₂ simulation and the control.

Difference in dissolved oxygen (μmol kg⁻¹) loss from solubility between the 4xCO₂ simulation and the control.

Difference in dissolved oxygen (μmol m³ yr⁻¹) consumption by remineralization between the 4xCO₂ simulation and the control.

Figure 11. Mechanisms for oxygen loss in the OMZs at 4 X $CO_2$. (a) Reference simulation. (b) The difference in DO concentrations between 4 X $CO_2$ and the reference simulation. (c) The ifference in DO lost due to changes in solubility between 4 X $CO_2$ and the reference simulation. (d) The increase in oxygen consumption due to remineralization of organic carbon between the 4 X $CO_2$ and reference simulation.